# Impacts of East Asian Summer and Winter Monsoon on Interannual Variations of Mass Concentrations and Direct Radiative Forcing of Black Carbon over Eastern China

**Y. H. Mao[1*], H. Liao[1,2], and H. S. Chen[2,3,4]**

[1]School of Environmental Science and Engineering, Nanjing University of Information Science and Technology (NUIST), Nanjing 210044, China

[2]International Joint Research Laboratory on Climate and Environment Change (ILCEC), NUIST, Nanjing 210044, China

[3]Collaborative Innovation Center on Forecast and Evaluation of Meteorological Disasters (CIC)/Key Laboratory of Meteorological Disaster, Ministry of Education (KLME), NUIST, Nanjing 210044, China

[4]School of Atmospheric Sciences, NUIST, Nanjing 210044, China

[*]*Corresponding author address:* Y. H. Mao (yhmao@nuist.edu.cn)

**Abstract.** We applied a global three-dimensional chemical transport model (GEOS-Chem) to examine the impacts of the East Asian monsoon on the interannual variations of mass concentrations and direct radiative forcing (DRF) of black carbon (BC) over eastern China (110–125 °E, 20–45 °N). With emissions fixed at the year 2010 levels, model simulations were driven by the Goddard Earth Observing System (GEOS-4) meteorological fields for 1986–2006 and the Modern Era Retrospective-analysis for Research and Applications (MERRA) meteorological fields for 1980–2010. During the period of 1986–2006, simulated JJA and DJF surface BC concentrations were higher in MERRA than in GEOS-4 by 0.30 µg m$^{-3}$ (44%) and 0.77 µg m$^{-3}$ (54%), respectively, because of the generally weaker precipitation in MERRA. We found that the strength of the East Asian summer monsoon (EASM, (East Asian winter monsoon, EAWM)) negatively correlated with simulated JJA (DJF) surface BC concentrations ($r = -0.7$ (–0.7) in GEOS-4 and –0.4 (–0.7) in MERRA), mainly by the changes in atmospheric circulation. Relative to the five strongest EASM years, simulated JJA surface BC concentrations in the five weakest monsoon years were higher over northern China (110–125 °E, 28–45 °N) by 0.04–0.09 µg m$^{-3}$ (3–11%), but lower over southern China (110–125 °E, 20–27 °N) by 0.03–0.04 µg m$^{-3}$ (10–11%). Compared to the five strongest EAWM years, simulated DJF surface BC concentrations in the five weakest monsoon years were higher by 0.13–0.15 µg m$^{-3}$ (5–8%) in northern China and by 0.04–0.10 µg m$^{-3}$ (3–12%) in southern China. The resulting JJA (DJF) mean all-sky DRF of BC at the top of the atmosphere were 0.04 W m$^{-2}$ (3%, (0.03 W m$^{-2}$, 2%)) higher in northern China but 0.06 W m$^{-2}$ (14%, (0.03 W m$^{-2}$, 3%)) lower in southern China. In the weakest monsoon years, the weaker vertical convection at the elevated altitudes led to the lower BC concentrations above 1–2 km in southern China, and therefore the lower BC DRF in the region. The differences in vertical profiles of BC between the weakest and strongest EASM years (1998–1997) and EAWM years (1990–1996) reached up to –0.09 µg m$^{-3}$ (–46%) and –0.08 µg m$^{-3}$ (–11%) at 1–2 km in eastern China.

# 1 Introduction

High concentrations of aerosols in China have been reported in recent years (e.g., Zhang et al., 2008, 2012), which are largely attributed to the increases in emissions due to the rapid economic development. In addition, studies have shown that meteorological parameters are important factors in driving the interannual variations of aerosols in China (e.g., Jeong and Park, 2013; Mu and Liao, 2014; Yang et al., 2015). For example, Mu and Liao (2014) reported that meteorological parameters, e.g., precipitation, wind direction and wind speed, and boundary layer condition, significantly influence the variations of emissions (biomass burning emissions), transport, and deposition of aerosols.

China is located in the East Asian monsoon (EAM) domain. In a strong (weak) summer monsoon year, China experiences strong (weak) southerlies, large rainfall in northern (southern) China, and a deficit of rainfall in the middle and lower reaches of the Yangtze River (northern China) (Zhu et al., 2012). A strong winter monsoon is characterized by a stronger Siberian High and Aleutian Low (Chen et al., 2000), and China thus experiences stronger northerlies, more active cold surge, lower surface temperature, and excess snowfall (Jhun and Lee, 2004). The EAM has been reported to influence the interannual variations of aerosols in China, via in changes in monsoon circulation, precipitation, vertical convection, and etc. (e.g., Liu et al., 2010; Zhang et al., 2010a, 2010b; Yan et al., 2011; Zhu et al., 2012). The observed weakening EAM in recently years is also considered to contribute to the increase in aerosols in eastern Asia (e.g., Chang et al., 2000; Ding et al., 2008; Wang et al., 2009; Zhou et al., 2015).

Studies have reported that the strength of the East Asian summer monsoon (EASM) negatively influences the interannual variations of aerosols in eastern China. Tan et al. (2015) showed that both the MODIS aerosol mass concentration and fine mode fraction in eastern China are high during weak monsoon years but low during active monsoon years for 2003–2013. By using the National Centers for Environmental Prediction/National Center for Atmospheric Research (NCEP/NCAR) reanalysis data and surface observations, Zhang et al. (2016) reported that the

frequency of occurrence of cyclone related weather patterns decreases in the weak EASM years, which significantly degrades the air quality in northern China for 1980–2013. Modeling studies also reported that the strength of the EASM influences simulated aerosol concentrations and optical depths over eastern Asia (Zhang et al., 2010a, 2010b; Yan et al., 2011; Zhu et al., 2012). For example, Zhu et al. (2012) using a global chemical transport model (GEOS-Chem) found that simulated summer surface $PM_{2.5}$ (particulate matter with a diameter of 2.5 μm or less) concentrations averaged over eastern China (110–125 ° E, 20–45 ° N) are ~18% higher in the five weakest summer monsoon years than in the five strongest monsoon years for 1986–2006.

Similarly, negative correlations have been found between the strength of the East Asian winter monsoon (EAWM) and changes of air quality in eastern China. By analyzing the observed visibility and meteorological parameters from surface stations, studies have shown that the weak EAWM is related to the decrease of cold wave occurrence and surface wind speed, and therefore partially accounts for the decrease of winter visibility and the increase of number of haze days and the severe haze pollution events in China from 1960s (Wang et al., 2014; Qu et al., 2015; Yin et al., 2015; Zhang et al., 2016). By further analyzing the reanalysis data, e.g., NCEP/NCAR and European Centre for Medium-Range Weather Forecasts (ECMWF), Li et al. (2015) showed that the stronger (weaker) EAWM is correlated with the less (more) wintertime fog–haze days. The weak EAWM results in a reduction of wind speed and decline in the frequency of northerly winds, which leads to an increase in the number of haze days and occurrences of severe haze events (Chen and Wang, 2015; Zhou et al., 2015).

Black carbon (BC) as a chemically inert species is a good tracer to investigate the impact of the meteorological parameters and the EAM on the interannual variations of aerosols. BC is an important short-lived aerosol; the reduction of BC emissions is identified as a near-term approach to benefit the human health, air quality, and climate change efficiently (Ramanathan and Xu, 2010; Shindell et al., 2012; Bond et al., 2013; IPCC, 2013; Smith et al., 2013). BC emissions in China have been dramatically

increased in the recent several decades, which contribute about 25% of the global total emissions (Cooke et al., 1999; Bond et al., 2004; Lu et al., 2011; Qin and Xie, 2012; Wang et al., 2012). Observed annual mean surface BC concentrations are typically about 2–5 μg m$^{-3}$ at rural sites (Zhang et al., 2008). Simulated annual direct radiative forcing (DRF) due to BC at the top of the atmosphere (TOA) is in the range of 0.58–1.46 W m$^{-2}$ in China, reported by previous modeling studies (summarized in Li et al., 2016). Mao et al. (2016) using the GEOS-Chem model showed that annual mean BC DRF averaged over China increases by 0.35 W m$^{-2}$ (51%) between 2010 and 1980.

The changes in BC concentrations in China are coupled with the changes in monsoon (e.g., Menon et al., 2002; Lau et al., 2006). Studies in the past decades were generally focused on the impacts of BC on the Asian monsoon (Menon et al., 2002; Lau et al., 2006; Meehl et al., 2007; Bollasina et al., 2011). Studies also showed that the climate effect of increasing BC could partially explain the "north drought/south flooding" precipitation pattern in China in recent decades (e.g., Menon et al., 2002; Gu et al., 2010). Conversely, the EAM could influence the spatial and vertical distributions of BC concentrations and further the radiative forcing and climate effect of BC. Zhu et al. (2012) showed that simulated summer surface BC concentrations averaged over northern China (110–125 $^\circ$ E, 28–45 $^\circ$ N) are ~11% higher in the five weakest monsoon years than in the five strongest monsoon years for 1986–2006. However, to our knowledge, few studies have systematically quantified the impact the EAM (especially the EAWM) on the variations of concentrations and DRF of BC in China.

The goal of the present study is to improve our understanding of the impacts of the EAM on the interannual variations of surface concentrations, vertical distributions, and DRF of BC in eastern China for 1986–2006. We aim to examine the mechanisms through which the EASM and EAWM influence the variations of BC. We describe the GEOS-Chem model and numerical simulations in Sect. 2. Sect. 3 shows simulated impacts of the EASM on interannual variations of June-July-August (JJA) BC in eastern China and examines the influence mechanisms. Sect. 4 presents the impacts of the EAWM on interannual variations of December-January-February (DJF) BC and

the relevant mechanisms. Summary and conclusions are given in Sect. 5.

## 2 Methods

### 2.1 GEOS-Chem Model and Numerical Experiments

The GEOS-Chem model is driven by assimilated meteorology from the Goddard Earth Observing System (GEOS) of the NASA Global Modeling and Assimilation Office (GMAO, Bey et al., 2001). Here we use GEOS-Chem version 9-01-03 (available at http://geos-chem.org) driven by the GEOS-4 and the Modern Era Retrospective-analysis for Research and Applications (MERRA) meteorological fields (Rienecker et al., 2011), with 6 h temporal resolution (3 h for surface variables and mixing depths), $2°$ (latitude) $\times 2.5°$ (longitude) horizontal resolution, and 30 (GEOS-4) or 47 (MERRA) vertical layers from the surface to 0.01 hPa. The GEOS-Chem simulation of carbonaceous aerosols has been reported previously by Park et al. (2003). Eighty percent of BC emitted from primary sources is assumed to be hydrophobic, and hydrophobic aerosols become hydrophilic with an e-folding time of 1.2 days (Cooke et al., 1999; Chin et al., 2002; Park et al., 2003). BC in the model is assumed to be externally mixed with other aerosol species.

Tracer advection is computed every 15 minutes with a flux-form semi-Lagrangian method (Lin and Rood, 1996). Tracer moist convection is computed using GEOS convective, entrainment, and detrainment mass fluxes as described by Allen et al. (1996a, b). The deep convection scheme of GEOS-4 is based on Zhang and McFarlane (1995), and the shallow convection treatment follows Hack (1994). MERRA convection is parameterized using the relaxed Arakawa-Schubert scheme (Arakawa and Schubert, 1974; Moorthi and Suarez, 1992). Simulation of aerosol wet and dry deposition follows Liu et al. (2001) and is updated by Wang et al. (2011). Wet deposition includes contributions from scavenging in convective updrafts, rainout from convective anvils, and rainout and washout from large-scale precipitation. Dry deposition of aerosols uses a resistance-in-series model (Walcek et al., 1986) dependent on local surface type and meteorological conditions.

The anthropogenic emissions of BC, including both fossil fuel and biofuel emissions, are from Bond et al. (2007) globally and updated in Asia (60 °E–150 °E, 10 ° S–55 ° N) with the Regional Emission inventory in Asia (REAS, available at http://www.jamstec.go.jp/frsgc/research/d4/emission.htm, Ohara et al., 2007). Seasonal variations of anthropogenic emissions are considered in China and Indian using monthly scaling factors taken from Kurokawa et al. (2013). Global biomass burning emissions of BC are taken from the Global Fire Emissions Database version 3 (GFEDv3, van der Werf et al., 2010) with a monthly temporal resolution. More details about the anthropogenic and biomass burning emissions of BC are discussed by Mao et al. (2016).

We conduct two simulations driven by GEOS-4 for years 1986–2006 (VMETG4) and by MERRA for 1980–2010 (VMET). Our analysis centers on the period of 1986–2006, the years for which both GEOS-4 and MERRA data are available. Both simulations are preceded by 1-year spin up. In each simulation, meteorological parameters are allowed to vary year to year, but anthropogenic and biomass burning emissions of BC are fixed at the year 2010 levels. The simulations thus represent the impact of variations in meteorological parameters on the interannual variations of BC. We also conduct simulation (VNOC) to quantify the contributions of the non-China emissions to BC. The configurations of the model simulation are the same as those in VMET, except that anthropogenic and biomass burning emissions in China are set to zero. The evaluations of GEOS-Chem aerosol simulations in China using the MERRA and GEOS-4 data are discussed in the studies, e.g., Mao et al. (2016) and Yang et al. (2015), respectively. In addition, we have systematically evaluated the BC simulations for 1980-2010 in China from the GEOS-Chem model (Li et al., 2016; Mao et al., 2016). We would like to point out that simulated BC concentrations are likely underestimated because of the biased low emissions (e.g., Bond et al., 2013; Xu et al., 2013; Mao et al., 2016) and coarse resolution of the model used. We have discussed the adjustment of the biased low BC emissions using the scaling factor in our previous study by Mao et al. (2016). The adjustment of the BC emissions is not included in the present study, as we aim to discuss the impact of variations in

meteorological parameters on BC.

**2.2 The Definition of EAM Index**

The interannual variations in the strength of the EAM are commonly represented by the indexes. Following Zhu et al. (2012) and Yang et al. (2014), we use the EASM index (EASMI, **Fig. 1a**) introduced by Li and Zeng (2002) in the present study based on the GEOS-4 meteorological parameters for 1986–2006 or the MERRA data for 1980–2010 (referred to as EASMI_GEOS and EASMI_MERRA, respectively). The EASMI calculated using the reanalyzed NCEP/NCAR datasets (Kalnay et al., 1996; Zhu et al., 2012, referred to as EASMI_NCEP, not shown) agrees well ($r > 0.97$) with EASMI_GEOS for 1986–2006 and with EASMI_MERRA for 1980–2010, indicating that both the GEOS-4 and MERRA data have a good representation of the strength of the EASM. Positive values of EASMI indicate strong summer monsoon years while negative values indicate weak monsoon years.

Numerous studies have shown that the intensity of the EAWM is closely tied with wind, air temperature, and precipitation (e.g., Guo et al., 1994; Ji et al., 1997; Chen et al., 2000; Jhun and Lee, 2004; Yan et al., 2009). The definitions of the EAWM index (EAWMI) are thus quite different in the previous studies (Table 1). Here we calculate the EAWMI (**Fig. 1b**) as the sum of zonal sea level pressure differences ($110\,^{\circ}$ E vs.$160\,^{\circ}$ E) over $20$–$70\,^{\circ}$ N, following Wu and Wang (2002). The EAWMIs in GEOS-4 and MERRA (referred to as EAWMI_GEOS and EAWMI_MERRA) in the present study show strong correlations with those based on surface temperature, wind, and pressure ($r = 0.51$–$0.82$, Table 1) and are generally consistent with that in NECP (referred to as EAWMI_NCEP), with the correlation coefficients larger than 0.94. The EAWMIs in GEOS and MERRA are thus reliable to represent the strength of the EAWM. Similarly, negative (positive) values of EAWMI indicate weak (strong) winter monsoon years.

## 3. Impact of EASM on Interannual Variation of BC

### 3.1 Simulated JJA BC in GEOS-4 and MERRA

**Fig. 1a** also show simulated JJA surface concentrations of BC averaged over eastern China (110–125 °E, 20–45 °N). Simulated JJA surface concentrations of BC have strong interannual variations, which range from 0.95–1.04 μg m$^{-3}$ with the deviation from the mean (DM) of −5.3% to 4.2% in VMET and 0.65–0.78 μg m$^{-3}$ with the DM of −6.8% to 12.5% in VMETG4. During the period of 1986–2006, JJA surface BC concentrations on average are 0.30 μg m$^{-3}$ (44%) higher in MERRA than in GEOS-4. Our analyses indicate that different precipitation patterns between GEOS-4 and MERRA likely account for the abovementioned differences in BC concentrations using the two meteorological fields.

We find that the JJA mean precipitation is stronger in GEOS-4 than in MERRA in most of China, except in southern China (**Fig. 1S**). In **Fig. 2a**, we further compare the differences in precipitation between GEOS-4 and MERRA averaged over eastern China. The JJA mean precipitation in GEOS-4 is 2.5 mm d$^{-1}$ (29%) stronger than that in MERRA for 1986–2006. The resulting wet deposition (**Fig. 2b**) is also higher by 0.018 kg s$^{-1}$ (11%) in GEOS-4 than in MERRA. The stronger precipitation in GEOS-4 thus results in the significantly lower surface BC concentrations. Note that MERRA is likely more reliable than the previous versions of GMAO metrological data products (e.g., GEOS-4 and GEOS-5), as MERRA has significant improved the convection and then precipitation and water vapor by comparing to the reanalyses (Rienecker et al., 2011).

### 3.2 Correlation between JJA BC and EASMI

In simulations VMET and VMETG4, we find that monsoon strength has large impacts on summertime BC concentrations over eastern China. JJA surface concentrations of BC negatively correlate with both the EASMI_GEOS4 and EASMI_MERRA (**Fig. 1a**). The correlation coefficient between simulated surface BC

concentrations and the EASMI_GEOS4 is –0.7 for 1986–2006, and those for the EASMI_MERRA are –0.5 for 1980–2010 and –0.4 for 1986–2006. Simulated surface BC concentrations are thus high (low) in the weak (strong) EASM years.

**Fig. 3a** shows the spatial distributions of the correlation coefficients between BC surface concentrations and the EASMI_GEOS4 or EASMI_MERRA. Negative correlations are found in central and northeastern China with the strongest negative correlations in eastern China and the Tibetan Plateau (<–0.8), while positive correlations are over southern and northwestern China with the largest values in southern China (> 0.7). The correlation coefficients in GEOS-4 and MERRA show similar spatial distribution and magnitude, except that positive correlations are found in larger regions in MERRA than in GEOS-4. Our results are generally consistent with those from Zhu et al. (2012), which reported that surface concentrations of $PM_{2.5}$ in GEOS-4 are high in northern China (110–125 °E, 28–45 °N) but low in southern China (110–125 °E, 20–27 °N) in the weak EASM years than in the strong monsoon years.

**3.3 Differences in BC between Weak and Strong EASM years**

In order to quantify to what degree the strength of the EASM influences surface BC concentrations in China, we examine the differences in the JJA mean surface BC concentrations between five weakest (1988, 1993, 1995, 1996, and 1998) and five strongest (1990, 1994, 1997, 2004, and 2006) EASM years during 1986–2006 (**Fig. 4a**). We select these weakest (or strongest) monsoon years based on the five largest negative (or positive) values of the normalized EASMI in both GEOS-4 and MERRA within 1986–2006. The selected monsoon years are thus slightly different with those from previous studies (Zhu et al., 2012; Yang et al. 2014) only based on GEOS-4 (weakest monsoon years (1988, 1989, 1996, 1998, and 2003), and strongest monsoon years (1990, 1994, 1997, 2002, and 2006)). The spatial distribution of the differences in concentrations between the weakest and strongest summer monsoon years is in good agreement with the distribution of the correlation coefficients between

concentrations and EASMI (**Fig. 3a**). The differences in JJA mean surface BC concentrations are highest in northern China with a maximum exceeding 0.3 μg m$^{-3}$ (40%). Relative to the strongest summer monsoon years, JJA surface BC concentrations in GEOS-4 in the weakest summer monsoon years are 0.09 μg m$^{-3}$ (11%) higher over northern China and 0.03 μg m$^{-3}$ (11%) lower over southern China (Table 2). The corresponding values in MERRA are 0.04 μg m$^{-3}$ (3%) higher over northern China and 0.04 μg m$^{-3}$ (10%) lower over southern China. In the eastern China, JJA surface BC concentrations in the weakest monsoon years are higher on average by 0.05 μg m$^{-3}$ (9%) in GEOS-4 and by 0.02 μg m$^{-3}$ (2%) in MERRA. The difference in surface BC concentrations between the weakest and strongest summer monsoon years in each region is comparable or even larger than the corresponding standard deviation of JJA mean surface BC for 1986–2006 (Table 2). The different patterns of BC concentrations between northern and southern China can also been see in **Fig. 5a**, which shows the height-latitude plot of the differences in BC concentrations averaged over 110–125 °E between the five weakest and five strongest monsoon years. BC concentrations in the whole troposphere are lower south of 27 °N but higher north of 27 °N in the weakest monsoon years than in the strongest years. The different patterns of BC concentrations between GEOS-4 and MERRA in Fig. 5a are likely because of the different convection schemes used in the two meteorological data (Rienecker et al., 2011).

Zhu et al. (2012) have shown that the impacts of the EASM on aerosol concentrations in eastern China are mainly by the changes in atmospheric circulation. **Fig. 6a** shows composite differences in JJA 850 hPa wind (m s$^{-1}$) between the five weakest and five strongest EASM years from the GEOS-4 and MERRA data. Relative to the strongest EASM years, anomalous northerlies over northern China and anomalous northeasterlies over the western North Pacific in the weakest monsoon years prevent the outflow of pollutants from northern China. In addition, southerly branch of the anomalous anticyclone in the south of the middle and lower reaches of the Yangtze River and nearby oceans strengthens the northward transport of aerosols from southern China to northern China. As a result, an anomalous convergence in

northern China leads to an increase in BC concentrations in the region, while an

anomalous anticyclone in the south of the middle and lower reaches of the Yangtze

River results in the decreased BC concentrations in southern China (**Fig. 4a**). The

convergence and divergence can also be seen in **Fig. 7a**, which shows anomalous

vertical transport of BC concentrations averaged over 110–125 $^\circ$ E. Compared to the

strong monsoon years, the increased surface BC concentrations in northern China lead

to higher upward mass fluxes of BC concentrations north of 25 $^\circ$ N in both MERRA

and GEOS-4. In southern China, the lower surface BC concentrations in the weakest

EASM years result in the decreased upward fluxes south of 25 $^\circ$ N. The pattern of the

anomalous vertical transport of BC concentrations thus confirms the anomalous

convergence in northern China and anomalous divergence in southern China in the

weakest monsoon years.

The differences in winds between the weak and strong monsoon years lead to

differences in horizontal transport of BC. We summarize in Table 3 the differences in

simulated horizontal mass fluxes of JJA BC at the four lateral boundaries of the box

in northern and southern China (**Fig. 4a**, from the surface to 10 km), based on

simulations VMETG4 and VMET. The boxes are selected as BC concentrations in the

regions are higher or lower in the weakest monsoon years than in the strongest

monsoon years (**Fig. 4a**). In northern China, the weakest (strongest) monsoon years in

GEOS-4 show inflow fluxes of BC by 2.24 (0.97) kg s$^{-1}$ at the south boundary and by

6.60 (4.20) kg s$^{-1}$ at the west boundary, and outflow fluxes of BC by 3.44 (4.06) kg

s$^{-1}$ at the north boundary and by 12.48 (9.20) kg s$^{-1}$ at the east boundary. The total

effects are thus outflow fluxes by 7.08 kg s$^{-1}$ in the weakest monsoon years and by

8.09 kg s$^{-1}$ in the strongest monsoon years, resulting in a net effect of larger inflow of

BC by 1.01 kg s$^{-1}$ in the weakest monsoon years than in the strongest monsoon years.

Similarly, simulation results in MERRA show a net effect of larger inflow of BC by

1.60 kg s$^{-1}$ in the weakest monsoon years than in the strongest monsoon years. The

larger inflow of BC in the weakest monsoon years thus leads to the higher surface BC

concentrations in northern China. In southern China, we find inflow fluxes of BC by

0.62 (0.70) kg s$^{-1}$ at the south boundary and by 0.94 (0.13) kg s$^{-1}$ at the west boundary,

and outflow fluxes of BC by 1.79 (0.88) kg s$^{-1}$ at the north boundary and by 0.33 (0.42) kg s$^{-1}$ at the east boundary in the weakest (strongest) monsoon years in GEOS-4. The resulting effect is larger outflow fluxes of BC by 0.09 kg s$^{-1}$ in the weakest monsoon years (0.56 kg s$^{-1}$) than in the strongest monsoon years (0.47 kg s$^{-1}$). In MERRA, the weakest monsoon years also show larger outflow fluxes of BC by 0.27 kg s$^{-1}$, compared to the strongest monsoon years. These results indicate that the differences in transport of BC due to the changes in atmospheric circulation are a dominant mechanism through which the EASM influences the variations of JJA BC concentrations in eastern China.

We also examine the impact of the changes in precipitation associated with the strength of the summer monsoon on BC concentrations, which is not as dominant as that of the winds. Compared to the strongest EASM years, increases in wet deposition of BC are found in the weakest monsoon years north of 28 °N in eastern China (Table 2), as a result of the high aerosol concentrations in the region and also the increased rainfall in the lower and middle reaches of the Yangtze River (around 30 °N). In the region south of 28 °N in eastern China, we find decreased wet deposition of BC in the weakest monsoon years because of the less rainfall and low BC concentrations in that region.

We would like to point out that warming trend is not a significant factor to the variations of BC concentrations in the present study, as emissions are fixed at the 2010 levels and warming trend in the emissions is thus excluded. In addition, Yang et al. (2016) have systematically examined the trends of metrological parameters and PM$_{2.5}$ in eastern China for 1985–2005. They found positive trend in temperature and negative trend in precipitation while no significant trend in BC concentrations.

**3.4 Impact of EASM on Vertical Profile and DRF of BC**

Previous studies have shown that vertical distribution of BC is critical for the calculation of the BC DRF (e.g., Bond et al., 2013; Li et al., 2016). The calculation of the BC DRF is dependent on several factors, e.g., BC lifetime and radiative forcing

efficiency (radiative forcing exerted per gram of BC), which are significantly influenced by vertical distribution of BC. Vertical profile of BC affects its wet scavenging and hence its lifetime (Bond et al., 2013). The direct radiative forcing efficiency of BC enhanced considerably when BC is located at high altitude largely because of the radiative interactions with clouds (Samset et al., 2013). For example, BC above 5 km accounts for ~40% of the global DRF of BC (Samset et al., 2013). We would like to point out that few aircraft observations of BC vertical profile are available in China. Previous studies have evaluated the GEOS-Chem simulated vertical profiles of BC by using datasets from aircraft campaigns for the regions of the Northwest Pacific, North America, and the Arctic (Park et al., 2005; Drury et al., 2010; Wang et al., 2011).

**Fig. 8a** compares the simulated JJA mean all-sky DRF of BC at the TOA in the five weakest and five strongest EASM years during 1986–2006. Model results are from simulation VMET. The BC DRF is calculated using the Rapid Radiative Transfer Model for GCMs (RRTMG, Heald et al., 2014), which is discussed in details by Mao et al. (2016). We find that the BC DRF is highest ($> 3.0$ W m$^{-2}$) over northern China in JJA. The spatial distributions of the differences in the BC DRF between the weakest and strongest monsoon years are similar to those in BC concentrations (**Fig. 4a**). Relative to the strongest monsoon years, the TOA DRF of BC shows an increase north of 28$^\circ$N while a reduction south of 27$^\circ$N in the weakest monsoon years. The BC DRF in northern China is 0.04 W m$^{-2}$ (3%, Table 4) higher in the weakest than strongest monsoon years, with a maximum of 0.3 W m$^{-2}$ in Jiangsu province. In southern China, the weakest monsoon years have a lower DRF by 0.06 W m$^{-2}$ (14%). As a result, the TOA DRF of BC in eastern China is 0.01 W m$^{-2}$ (1%) higher in the weakest monsoon years than in the strongest monsoon years. Note that the estimated DRF is associated with large uncertainties due to the BC mixing state used in model, which assumes external mixing of aerosols and gives a lower-bound estimate of BC DRF. Internal mixing of BC with scattering aerosols in the real atmosphere likely increases the estimates of DRF (e.g., Jacobson, 2001).

We further compare in **Fig. 9a** the vertical distribution of simulated JJA mean

all-sky DRF of BC in the five weakest and five strongest EASM years, averaged over 110–125 $°$E. We find largest BC-induced forcing at the latitude of 30–40 $°$N in the weakest monsoon years and 35–40 $°$N in the strongest monsoon years. The shift of the center of the highest BC DRF is likely due to the different vertical distributions of BC concentrations between the weakest and strongest monsoon years (**Fig. 5a**). BC DRF is higher by >0.13 W m$^{-2}$ (10–20%) over 30–35 $°$N in the five weakest EASM years compared to the five strongest EASM years, which are consistently with those in **Fig. 8a**. A maximum BC DRF (>2 W m$^{-2}$) is shown approximately at an altitude of 3–10 km, because of the larger direct radiative forcing efficiency of BC at high altitude.

**Fig. 10a** shows the simulated vertical profiles of JJA BC mass concentrations (μg m$^{-3}$) averaged over eastern China for 1986–2006. The simulated BC concentrations are higher in MERRA than in GEOS-4 below 3 km. We find that the vertical profiles of JJA BC in GEOS-4 generally show larger interannual variations than those in MERRA. The variations of JJA BC in MERRA and in GEOS-4 range from –5% to 4% (–7% to 12%) at the surface, –25% to 16% (–23% to 23%) at 1 km, –35% to 42% (–32% to 46%) at 2 km, –23% to 32% (–25% to 67%) at 3 km, –13% to 10% (–18% to 71%) at 4 km, –10% to 7% (–14% to >76%) at 5–8 km. The differences in vertical profiles of BC in MERRA between the weakest and strongest EASM years (1998–1997) are –46% to 7%, with the largest differences of –0.09 μg m$^{-3}$ at ~2 km. We further compare the differences in simulated vertical profiles of JJA BC between the five weakest and five strongest EASM years averaged over northern and southern China in MERRA. The decreased BC concentrations throughout the troposphere in the weakest monsoon years lead to a reduction in the BC DRF in southern China (Table 4), while the increased BC concentrations below 2 km result in a significant increase of the BC DRF in northern China (Table 4).

Studies have shown that the impact of non-China emissions is significant on vertical profiles and hence DRF of BC in China; the contributions of non-China emissions to concentrations and DRF of BC in China are larger than 20% at 5 km altitude and about 17–43%, respectively (e.g., Li et al., 2016). **Figure 11a** shows vertical distribution of simulated JJA mean all-sky DRF of BC due to non-China

emissions in the five weakest and five strongest EASM years, averaged over 110–125 °E. Model results are from simulation VNOC, in which the anthropogenic and biomass burning emissions are turned off in China. The non-China emissions induce a high (> 0.16 W m$^{-2}$) BC DRF above ~5 km due to the significant contributions of non-China emissions to BC concentrations at high altitudes. Compared to the five strongest EASM years, the simulated DRF of BC due to non-China emissions in the weakest EASM years is larger (by ~10%) at 25–40 °N, because of the higher (by > 10%) BC concentrations transported to the region (**Fig. 12a**).

## 4 Impact of EAWM on Interannual Variation of BC

### 4.1 Simulated DJF BC in GEOS-4 and MERRA

Simulated DJF surface BC concentrations averaged over eastern China also have strong interannual variations, ranging from 1.30–1.58 μg m$^{-3}$ (–8.9 to 10.8%) in GEOS-4 for 1986–2006 and from 2.05–2.31 μg m$^{-3}$ (–7.0% to 5.2%) in MERRA for 1980–2010 (**Fig. 1b**). DJF mean surface concentrations of BC for 1986–2006 are 0.77 μg m$^{-3}$ (54%) higher in MERRA than in GEOS-4. Again, the consistently stronger precipitation in GEOS-4 (by 0.3 mm d$^{-1}$, 21% on average) largely accounts for the lower surface BC concentrations (**Figs. 1S** and **2a**). The DJF mean precipitation averaged for 1986–2006 is higher in GEOS-4 than in MERRA in most of China (**Fig. 1S**), except in the delta of the Yangtze River in eastern China. The resulting differences in BC wet deposition between GEOS-4 and MERRA show similar patterns as those in precipitation (not shown). The DJF mean wet deposition of BC in GEOS-4 is generally higher (by 0.007 kg s$^{-1}$, 5% on average) than that in MERRA for 1986–2006, except in 1998 (**Fig. 2b**). In addition, we find that the planetary boundary layer height (PBLH) partially accounts for the abovementioned differences in surface BC concentrations between GEOS-4 and MERRA. The DJF mean PBLH is generally higher in GEOS-4 than in MERRA by 11.6 m (2%, **Fig. S2**). The lower PBLH in MERRA suppresses the convection and thus leads to higher BC concentrations in the surface

## 4.2 Correlation between DJF BC and EAWMI

**Fig. 1b** shows the normalized EAWMI and simulated DJF mean surface BC concentrations averaged over eastern China from simulation VMET for 1980–2010 and from VMETG4 for 1986–2006. The correlation coefficient between the surface BC concentrations and the EAWMI_GEOS4 is –0.7 for 1986–2006, and those between surface BC and the EAWMI_MERRA are –0.6 and –0.7, respectively, for 1980–2010 and for 1986–2006. Different definitions of the EAWMI also show negative correlations with simulated DJF surface BC concentrations (Table 1, $r = -0.16$ to –0.72). This negative correlation between simulated DJF mean surface BC concentrations and the EAWMIs over eastern China indicates that surface BC concentrations are generally high in the weak winter monsoon years. The correlation coefficients in GEOS-4 and MERRA show similar spatial distribution and magnitude; negative correlations are found in most of China, while positive correlations are over southwestern China (**Fig. 3b**).

## 4.3 Differences in BC between Weak and Strong EAWM years

**Fig. 4b** shows the differences in simulated DJF mean surface BC concentrations ($\mu g \ m^{-3}$) between weakest (1990, 1993, 1997, 1998, and 2002) and strongest (1986, 1996, 2001, 2005, and 2006) EAWM years during 1986–2006 from model simulations using the GEOS-4 and MERRA data. The spatial distribution of the differences in concentrations is in good agreement with the distribution of the correlation coefficients between the EAWMI and surface BC (**Fig. 3b)**. In eastern China, DJF surface BC concentrations in GEOS-4 are 0.12 $\mu g \ m^{-3}$ (9%) higher in the weakest winter monsoon years than in the strongest years (Table 2). The corresponding values are 0.11 $\mu g \ m^{-3}$ (5%) higher in MERRA. In northern China, simulated surface BC concentrations are higher in the weakest monsoon years than in the strongest monsoon year by 0.13 $\mu g \ m^{-3}$ (8%) in GEOS-4 and by 0.14 $\mu g \ m^{-3}$ (5%) in MERRA. In southern China, the corresponding concentrations are higher by 0.10 $\mu g \ m^{-3}$ (12%) and 0.04 $\mu g \ m^{-3}$ (3%), respectively, in GEOS-4 and in MERRA. The

difference in surface BC concentrations between the weakest and strongest winter

monsoon years over each region is significant by comparing with the corresponding

mean and standard deviation of DJF mean surface BC for 1986–2006 (Table 2).We

find that the region over 30–40 $^\circ$ N has lower BC concentrations in the weakest

monsoon years. This lower concentrations are also shown in **Fig. 5b**, which represents

the height-latitude of differences in simulated DJF mean BC concentrations between

the five weakest and five strongest EAWM years during 1986–2006 and averaged

over 110–125 $^\circ$ E from model simulations VMETG4 and VMET. Increased BC

concentrations in the weakest monsoon years are found over north of 20 $^\circ$ N in both

GEOS-4 and MERRA, except the region over 30–40 $^\circ$N and above 1 km.

The changes in atmospheric circulation again likely account for the increased BC

concentrations in the weak winter monsoon years in eastern China. **Fig. 6b** shows the

composite differences in DJF 850 hPa wind (m s$^{-1}$) between the five weakest and five

strongest EAWM years from the GEOS-4 and MERRA data. The differences in wind

in GEOS-4 show a similar pattern as those in MERRA. In DJF, northerly winds are

weaker in the weaker monsoon years than in the stronger monsoon years. As a result,

anomalous southwesterlies are found in the weakest monsoon years along the coast of

eastern China and anomalies southeasterlies control northern China and northeast

China, which do not favor the outflow of pollutants from eastern China (Table 3). Fig.

**7b** shows the differences in simulated upward mass flux of DJF BC (kg s$^{-1}$) between

the five weakest and five strongest EAWM years. The differences are averaged over

the longitude range of 110–125 $^\circ$ E. Compared to the strongest monsoon years,

increases in upward mass flux of BC concentrations are found over 20–30 $^\circ$ N and

north of 40 $^\circ$ N in the troposphere in the weakest monsoon years, confirming the

increased surface BC concentrations in northern and southern China (**Figs. 4b** and **5b**).

We find decreased upward transport of BC over 30–40 $^\circ$N in the weakest monsoon

years, which is consistent with decreased concentrations in the region of static winds

(**Fig. 6b**). Our results are consistent with the studies, e.g., Li et al. (2015) and Zhou et

al. (2015), which showed that the change in wind speed and wind direction is the

major factor of the negative correlation between the increased winter fog–haze days

and the weaken of the EAWM in China.

We further summarize in Table 3 the differences in horizontal fluxes of DJF BC at the four lateral boundaries of the northern and southern boxes (**Fig. 4b**, from the surface to 10 km) between the five weakest and five strongest EAWM years, based on simulations VMETG4 and VMET. Both northern and southern China show increased BC concentrations in the weakest monsoon years than in the strongest monsoon years (**Fig. 4b**). In the southern box, we find larger inflow of BC by 1.67 (0.99) kg s$^{-1}$ at the west boundary, less inflow by 1.45 (1.19) kg s$^{-1}$ at north boundary, less outflow by 0.52 (0.70) kg s$^{-1}$ at the south boundary, and larger outflow by 0.55 (0.10) kg s$^{-1}$ at east boundary, from simulation VMETG4 (VMET). The net effect in southern China is a larger inflow of BC by 0.19 (0.40) kg s$^{-1}$ in the weakest EAWM years than in the strongest EAWM years, which leads to the higher BC concentrations in the weakest EAWM years. As we discussed in Sect. 3.3, the larger outflow fluxes of BC by 0.09 (0.27) kg s$^{-1}$ result in the lower BC concentrations in southern China in the weakest EASM years than in the strongest EASM years. Different patterns of atmospheric circulation between summer and winter monsoon thus lead to the different distributions of BC in southern China. In northern China, there is a net effect of larger inflow of BC by 0.64 (0.62) kg s$^{-1}$ because of the anomalous southerlies and westerlies in the weakest monsoon years. The anomalous southerlies in northern China thus prevent the outflow of pollutants and lead to an increase in BC concentrations in the region in the weakest monsoon years. Compared to the strongest EAWM years, enhanced wet deposition of BC are found in the weakest monsoon years in both northern and southern China (Table 2), likely because of the increased BC concentrations and precipitation in the corresponding regions. Weaker upward transport in the weakest monsoon years than the strongest years above 1-2 km in southern China (Fig. **7b**) is also not a dominate factor that contributes to the higher surface BC concentrations in the region (Tables 2 and 3).

**4.4 Impact of EAWM on Vertical Profile and DRF of BC**

**Fig. 8b** shows the simulated DJF mean all-sky TOA DRF of BC in the five weakest and strongest EAWM years during 1986–2006, based on simulation VMET. The simulated BC DRF is high in eastern China, with the largest values ($> 5.0$ W m$^{-2}$) in the Sichuan Basin. In northern China, the TOA DRF of BC is 0.03 W m$^{-2}$ (2%, Table 4) higher in the weakest monsoon years than in the strongest monsoon years, consistent to the higher BC concentrations in the region (**Fig. 4b**). We further separate northern China into two regions, the central China Plain (110–125 °E, 28–36 °N) and the northern China Plain (110–125 ° E, 37–45 ° N). Relative to the five strongest monsoon years, the BC DRF in the weakest monsoon years is higher in the northern China Plain by 0.11 W m$^{-2}$ (11%) but lower in the central China Plain by 0.03 W m$^{-2}$ (1%). In the central China Plain, although the surface concentrations are higher by 0.08 μg m$^{-3}$ (2%) in the weakest monsoon years, the corresponding DRF is lower partially because of the lower column burdens of tropospheric BC (by 0.04 mg m$^{-2}$, 1%, from surface to 10 km, Figs. 5(b right) and 10(b2)). In southern China, the DRF is 0.03 W m$^{-2}$ (3%) lower in the weakest monsoon years than in the strongest monsoon years. In contrast, both surface concentrations (higher by 0.04 μg m$^{-3}$, 3%) and column burdens (higher by 0.02 mg m$^{-2}$, 2%) of BC are higher in the weakest monsoon years. We attribute these discrepancies largely to the vertical distributions of BC concentrations as discussed in the following paragraph. We further compare in **Fig. 9b** the vertical distribution of simulated DJF DRF of BC in the five weakest and five strongest EAWM years, averaged over 110–125 °E. The BC-induced forcing is large ($>2.8$ W m$^{-2}$) at the latitude of 20–40 °N and at an altitude of 5–10 km. BC DRF is higher by $> 0.1$ W m$^{-2}$ ($> 10\%$) north of 35 °N in the five weakest EAWM years than in the five strongest EAWM years, consistent with those in **Fig. 8b**.

The abovementioned differences in spatial patterns of DRF and BC concentrations are likely because of the vertical distributions of BC concentrations. In general, the simulated vertical profiles of DJF BC concentrations are higher in MERRA than in GEOS-4, but the interannual variations are larger in GEOS-4 than in MERRA (**Fig. 10b**). The variations of DJF BC in MERRA (GEOS-4) range from −7% to 5% (−9% to 11%) at the surface, −12% to 10% (−13% to 27%) at 1 km, −19% to 14% (−13% to

62%) at 2 km, −14% to 15% (−17% to 57%) at 3 km, −17% to 16% (−22% to 61%) at 4 km, −17% to >14% (−22% to >67%) at 5–8 km. We find that the differences in vertical profiles of BC in MERRA between the weakest and strongest EAWM years (1990–1996) are −0.08 to 0.2 μg m$^{-3}$ (−11% to 12%) below 10 km, with the largest differences at the surface and ~1.5 km. We further compare the differences in simulated vertical profiles of DJF BC mass concentrations between the five weakest and five strongest EAWM years from model simulation VMET, averaged over southern China, the central China plain, and the northern China Plain. Relative to the strongest monsoon years, decreased BC concentrations are found in the weakest monsoon years from 2 to 5 km in southern China and from 1 to 6 km in the central China Plain. The decreased BC concentrations above 1–2 km lead to the reduction in the DRF in the two regions. In contrast, the higher DRF of BC in the northern China Plain in the weakest monsoon years is because of the increased BC concentrations throughout the troposphere.

The lower concentrations above 1–2 km in the weakest monsoon years in southern China and the central Chin Plain are likely because of the weaker vertical convection at the corresponding altitudes in the weakest monsoon years than in the strongest monsoon years. We calculate the horizontal and vertical fluxes of BC in two boxes of southern China and the central China Plain from 1 to 6 km (Table 5). In vertical direction, the two boxes have upward fluxes in both lower and upper boundaries. Relative to the strongest monsoon years, the southern box has a net outflow of 0.07 kg s$^{-1}$ in the weakest monsoon years; the central China Plain shows a net downward flux of 0.11 kg s$^{-1}$. The corresponding net horizontal fluxes are relatively smaller, and about 0.03 kg s$^{-1}$ in southern China and 0.01 kg s$^{-1}$ in the central China Plain. The weaker vertical fluxes above 1–2 km in the weakest monsoon years thus result in the lower BC concentrations at the elevated altitudes therefore the reduction in the DRF in the two regions.

**Fig. 11b** shows the vertical distribution of simulated DJF mean all-sky DRF of BC due to non-China emissions in the five weakest and five strongest EAWM years,

averaged over 110–125 °E. The non-China emissions induce a high (> 0.35 W m$^{-2}$) BC DRF at 15–35 °N. We also find a higher (by >5%) DRF of BC north of 25 °N in the weakest EAWM years than in the strongest years, due to the larger BC concentrations at the low troposphere in the weakest EAWM years (**Fig. 12b**).

## 5. Summary and conclusions

We quantified the impacts of the EASM and EAWM on the interannual variations of mass concentrations and DRF of BC in eastern China for 1986–2006 and examined the relevant mechanisms. We conducted simulations with fixed anthropogenic and biomass burning emissions at the year 2010 levels and driven by GEOS-4 for 1986–2006 and by MERRA for 1980–2010.

We found that simulated JJA and DJF surface BC concentrations averaged over eastern China were higher in MERRA than in GEOS-4 by 0.30 μg m$^{-3}$ (44%) and 0.77 μg m$^{-3}$ (54%), respectively. Our analyses indicated that generally higher precipitation in GEOS-4 than in MERRA largely accounted for the differences in BC concentrations using the two meteorological fields.

In JJA, simulated BC concentrations showed interannual variations of –5% to 4% in MERRA (–7% to 12% in GEOS-4) at the surface and –35% to 42% in MERRA (–32% to >76% in GEOS-4) above 1 km. The differences in vertical profiles of BC between the weakest and strongest EASM years (1998–1997) reached up to –0.09 μg m$^{-3}$ (–46%) at 1–2 km. Simulated JJA surface BC concentrations negatively correlated with the strength of the EASM ($r = -0.7$ in GEOS-4 and –0.4 in MERRA), mainly by the changes in atmospheric circulation. Relative to the five strongest EASM years, simulated JJA surface BC concentrations in the five weakest EASM years were higher over northern China by 0.09 μg m$^{-3}$ (11%) in GEOS-4 and by 0.04 μg m$^{-3}$ (3%) in MERRA. The corresponding concentrations were lower over southern China by 0.03 μg m$^{-3}$ (11%) and 0.04 μg m$^{-3}$ (10%). The resulting JJA mean TOA DRF of BC were 0.04 W m$^{-2}$ (3%) higher in northern China but 0.06 W m$^{-2}$ (14%)

lower in southern China.

In DJF, the changes in meteorological parameters alone led to interannual

variations in BC concentrations ranging from –7% to 5% in MERRA (–9% to 11% in

GEOS-4) at the surface and –19% to >14% in MERRA (–22% to >67% in GEOS-4)

above 1 km. Simulated DJF surface BC concentrations negatively correlated with the

EAWMI ($r$ = –0.7 in GEOS-4 and –0.7 in MERRA), indicating higher DJF surface

BC concentrations in the weaker EAWM years. We also found that the changes in

atmospheric circulation likely accounted for the increased BC concentrations in the

weak EAWM years. In winter, anomalous southerlies in the weak monsoon years did

not favor the outflow of pollutants, leading to an increase in BC concentration.

Compared to the five strongest EAWM years, simulated DJF surface BC

concentrations in the five weakest EAWM years were higher in northern China by

0.13 μg m$^{-3}$ (8%) in GEOS-4 and 0.14 μg m$^{-3}$ (5%) in MERRA. The corresponding

concentrations were also higher in southern China by 0.10 μg m$^{-3}$ (12%) and 0.04 μg

m$^{-3}$ (3%). The resulting TOA DRF of DJF BC was 0.03 W m$^{-2}$ (2%) higher in

northern China but 0.03 W m$^{-2}$ (2%) lower in southern China. In southern China, the

decreased BC concentrations above 1–2 km in the weakest EAWM years led to the

reduction in BC DRF, likely due to the weaker vertical convection in the

corresponding altitudes. The vertical profiles of BC are lower in weakest EAWM year

(1990) than in the strongest year (1996) above 1–2 km, with the largest values of –

0.08 μg m$^{-3}$ (–11%) in eastern China.

Different patterns of atmospheric circulation between summer and winter

monsoon lead to the different distributions of BC in southern and northern China.

Note that these different changes in BC concentrations and DRF between northern

and southern China due to the EAM would be useful for proposing efficient air

quality regulation in different regions of China. It is also worth to point out that the

BC DRF is also dependent on factors such as cloud and background aerosol

distributions (Samset et al., 2011), which can be influenced by the strength of the

EAM (Liu et al., 2010; Zhu et al., 2012). In addition, the strength of the EAWM

would influence the following summer monsoon via changes in the factors such as

circulation and precipitation (e.g., Chen et al., 2000), and further affect the aerosols concentrations and radiative forcing. These aspects should be further investigated in future studies.

*Acknowledgements.* This work was supported by the National Basic Research Program of China (973 program, Grant 2014CB441202), the Strategic Priority Research Program of the Chinese Academy of Sciences Strategic Priority Research Program Grant No. XDA05100503, the National Natural Science Foundation of China under grants 91544219, 41475137, and 41321064. The GEOS-Chem model is managed by the Atmospheric Chemistry Modeling group at Harvard University with support from the NASA ACMAP program.

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

**Tables and Figures**

**Table 1.** Correlation coefficients among different definitions of the strength of the East Asian winter monsoon (EAWM), and between the EAWM Index (EAWMI) and simulated December-January-February (DJF) mean surface BC concentrations averaged over eastern China (110–125 ° E, 20–45 ° N). Simulated BC concentrations are from model simulations VMETG4 and VMET, and corresponding monsoon indexes are calculated based on GEOS-4 and MERRA assimilated meteorological data.

**Table 2**. Simulated JJA (DJF) mean surface BC concentrations ($\mu$g m$^{-3}$) in the five weakest and five strongest EASM (EAWM) years during 1986–2006. Results are from simulations VMETG4 and VMET averaged over northern China (NC, 110–125 °E, 28–45 °N), southern China (SC, 110–125 °E, 20–27 ° N), and eastern China (EC, 110–125 °E, 20–45 °N).

**Table 3**. The composite analyses of JJA (DJF) horizontal fluxes of BC (kg s$^{-1}$) for two selected boxes (northern China (110–125 ° E, 28–45 ° N) and southern China (110–125 ° E, 20–27 ° N), from the surface to 10 km) based on simulations VMETG4 and VMET. The values are averages over the five weakest and five strongest EASM (EAWM) years during 1986–2006. For horizontal fluxes, positive values indicate eastward or northward transport and negative values indicate westward or southward transport. Also shown are the corresponding wet deposition of BC (kg s$^{-1}$) for the two selected boxes.

**Table 4**. Simulated JJA (DJF) mean all-sky direct radiative forcing (DRF) of BC (W m$^{-2}$) at the top of the atmosphere (TOA) in the five weakest and five strongest EASM (EAWM) years during 1986–2006. Results are from simulation VMET averaged over eastern China (110–125 ° E, 20–45 ° N),

northern China (110–125 °E, 28–45 °N), the northern China Plain (110–125 °E, 37–45 °N), the central China Plain (110–125 °E, 28–36 °N), and southern China (110–125 °E, 20–27 °N).

**Table 5**. The composite analyses of DJF horizontal and vertical fluxes of BC (kg s$^{-1}$) for two selected boxes (the central China Plain (110–125 °E, 27–36 °N) and southern China (110–125 ° E, 20–27 ° N), from 1 to 6 km) based on simulation VMET. The values are averages over the five weakest and five strongest EAWM years during 1986–2006. For fluxes, positive values indicate eastward, northward, or upward transport and negative values indicate westward, southward, or downward transport.

**Figure 1. (a)** Normalized East Asian summer monsoon Index (EASMI, bars, left y axis) and the simulated June-July-August (JJA) mean surface BC concentrations (lines, right y axis, µg m$^{-3}$) averaged over eastern China (20–45 °N, 110–125 °E) from model simulation VMET (red line) for 1980–2010 and from VMETG4 (blue line) for 1986–2006. EASMI are calculated based on MERRA (red bars) and GEOS-4 (blue bars) assimilated meteorological data following Li and Zeng (2002). **(b)** Same as **(a)**, but for normalized East Asian winter monsoon Index (EAWMI) and the simulated December-January-February (DJF) mean surface BC concentrations. EAWMIs are calculated following Wu and Wang (2002).

**Figure 2. (a)** JJA and DJF mean precipitation (mm d$^{-1}$) averaged over eastern China for 1986–2006 from GEOS-4 (blue lines) and MERRA (red lines) meteorological data. DJF mean precipitation is multiplied by 5 in **(a2)**. **(b)** Same as **(a)**, but for wet deposition (kg s$^{-1}$).

**Figure 3. (a)** Correlation coefficients between EASMI and JJA mean surface BC concentrations during 1986–2006. **(b)** Correlation coefficients between EAWMI and DJF mean surface BC concentrations during 1986–2006. Simulated BC concentrations are from model simulations VMETG4 (left) and VMET (right), and monsoon indexes are calculated based on GEOS-4 (left) and MERRA (right) assimilated meteorological data. The dotted areas indicate statistical significance with 95% confidence from a two-tailed Student's t test.

**Figure 4. (a1)** Absolute (µg m$^{-3}$) and **(a2)** percentage (%) differences in simulated JJA mean surface BC concentrations between weakest (1988, 1993, 1995, 1996, and 1998) and strongest (1990, 1994, 1997, 2004, and 2006) EASM years during 1986–2006 from model simulations VMETG4 and VMET. **(b1)** and **(b2)** Same as **(a1)** and **(a2)**, respectively, but for absolute (µg m$^{-3}$) and percentage (%) differences in simulated DJF mean surface BC concentrations between weakest (1990, 1993, 1997, 1998, and 2002) and strongest (1986, 1996, 2001, 2005, and 2006) EAWM years. The enclosed

areas are defined as northern China (NC, 110–125 ° E, 28–45 ° N) and southern China (SC, 110–125 °E, 20–27 °N).

**Figure 5. (a)** Height-latitude cross section of differences in simulated JJA mean BC concentrations ($\mu$g m$^{-3}$) between the five weakest and five strongest EASM years during 1986–2006. Plots are averaged over longitude range of 110–125 ° E from model simulations VMETG4 (left) and VMET (right). **(b)** Same as **(a)**, but for differences in DJF between five weakest and five strongest EAWM years.

**Figure 6. (a)** Differences in JJA 850 hPa wind (vector, m s$^{-1}$) between the five weakest and five strongest EASM years during 1986–2006 from GEOS-4 (left) and MERRA (right) data. **(b)** Same as **(a)**, but for differences in DJF wind between five weakest and five strongest EAWM years.

**Figure 7. (a)** Differences in simulated upward mass flux of JJA BC (kg s$^{-1}$) between the five weakest and five strongest EASM years during 1986–2006. Plots are averaged over longitude range of 110–125 ° E from model simulations VMETG4 (left) and VMET (right). **(b)** Same as **(a)**, but for differences in DJF between five weakest and five strongest EAWM years.

**Figure 8. (a)** Simulated JJA mean all-sky direct radiative forcing (DRF) of BC (W m$^{-2}$) at the top of the atmosphere (TOA) in the **(a1)** five weakest and **(a2)** five strongest EASM years during 1986–2006 from model simulation VMET. Also shown are the **(a3)** absolute (W m$^{-2}$) and **(a4)** percentage (%) differences between the five weakest and five strongest EASM years. **(b)** Same as **(a)**, but for simulated DJF mean all-sky TOA DRF of BC in the five weakest and five strongest EAWM years. The enclosed areas are defined as northern China (NC, 110–125 °E, 28–45 °N), the northern China Plain (NCP, 110–125 °E, 36–45 °N), the central China Plain (CCP, 110–125 °E, 28–36 °N), and southern China (SC, 110–125 °E, 20–27 °N).

**Figure 9. (a)** Height-latitude cross sections of simulated JJA mean all-sky DRF of BC (W m$^{-2}$) in the **(a1)** five weakest and **(a2)** five strongest EASM years during 1986–2006. Also shown are the **(a3)** absolute (W m$^{-2}$) and **(a4)** percentage (%) differences between the five weakest and five strongest EASM years. Plots are averaged over longitude range of 110–125 ° E from model simulation VMET. **(b)** Same as **(a)**, but for simulated DJF mean all-sky DRF of BC in the five weakest and five strongest EAWM years.

**Figure 10. (a1)** Simulated vertical profiles of JJA BC mass concentrations ($\mu$g m$^{-3}$) averaged over 1986–2006. The error bars represent the minimum and maximum values of BC. Results are averages over eastern China from model simulations VMETG4 (blue) and VMET (red). **(a2)** Differences in simulated vertical profiles of JJA BC mass concentrations ($\mu$g m$^{-3}$) between the five weakest and five strongest EAM years (solid lines) during 1986–

2006, and between the weakest and strongest EASM years (1998–1997, dotted lines). Results are averages over eastern China, northern China, and southern China from model simulations VMET. **(b1)** Same as **(a1),** but for simulated DJF BC mass concentrations. **(b2)** Same as **(a2),** but for differences in DJF between the five weakest and five strongest EAWM years and between the weakest and strongest EAWM years (1990–1996). Results are averages over eastern China, northern China Plain, the central China Plain, and southern China.

**Figure 11.** Same as **Figure 9,** but for the contributions from non-China emissions to simulated all-sky DRF of BC.

**Figure 12. (a)** Height-latitude cross sections of contributions of non-China emissions to simulated JJA mean BC concentrations ($\mu g \ m^{-3}$) in the **(a1)** five weakest and **(a2)** five strongest EASM years during 1986–2006. Also shown are the **(a3)** absolute ($\mu g \ m^{-3}$) and **(a4)** percentage (%) differences between the five weakest and five strongest EASM years. Plots are averaged over longitude range of 110–125 °E from model simulation VMET. **(b)** Same as **(a)**, but for simulated DJF mean BC concentrations in the five weakest and five strongest EAWM years

**Figure 1S.** JJA and DJF mean precipitation (mm $d^{-1}$) averaged for 1986–2006 from GEOS-4 **(a)** and MERRA **(b)** meteorological data. Also shown are the differences between GEOS-4 and MERRA data **(c)**.

**Figure 2S. (left)** Differences in DJF mean planetary boundary layer height (PBLH, m) averaged for 1986–2006 between GEOS-4 and MERRA. **(right)** DJF mean PBLH averaged over eastern China for 1986–2006 from GEOS-4 (blue line) and MERRA (red line).

**Table 1.** Correlation coefficients among different definitions of the strength of the

East Asian winter monsoon (EAWM), and between the EAWM Index (EAWMI) and

simulated December-January-February (DJF) mean surface BC concentrations

averaged over eastern China (110–125 $^\circ$E, 20–45 $^\circ$N). Simulated BC concentrations

are from model simulations VMETG4 and VMET, and corresponding monsoon

indexes are calculated based on GEOS-4 and MERRA assimilated meteorological

data.

| Correlation | GEOS–4 (1986-2006) | | MERRA (1986-2006) | | MERRA (1980-2010) | |
|---|---|---|---|---|---|---|
| | EAWMI[1] | BC | EAWMI | BC | EAWMI | BC |
| EAWMI_$T$[2] | 0.63 | –0.57 | 0.58 | –0.16 | 0.56 | –0.29 |
| EAWMI_$V$[3] | 0.51 | –0.31 | 0.56 | –0.50 | 0.54 | –0.40 |
| EAWMI_$U$[4] | 0.77 | –0.42 | 0.82 | –0.72 | 0.73 | –0.69 |
| EAWMI_$P_1$[5] | 0.65 | –0.33 | 0.72 | –0.38 | 0.77 | –0.41 |
| EAWMI_$P_2$[6] | 0.71 | –0.61 | 0.72 | –0.68 | 0.70 | –0.66 |

[1]$\text{EAWMI}_i = \text{norm}(\sum_{20^\circ\text{N}}^{70^\circ\text{N}}(P_{1i} - P_{2i}))$, $P_{1i}$ is the DJF mean sea level pressure over 110 $^\circ$E, $P_{2i}$ is the

DJF mean sea level pressure over 160 $^\circ$E (Wu and Wang, 2002).

[2]$\text{EAWMI}\_T_i = \bar{\bar{T}} - \overline{T_i}$, $\overline{T_i}$ is the DJF mean surface temperature over the region of 20–40 $^\circ$N and 110–

135 $^\circ$E for year $i$, $\bar{\bar{T}}$ is the mean of $\overline{T_i}$ (Yan et al., 2009).

[3]$\text{EAWMI}\_V_i = \bar{\bar{V}} - \overline{V_i}$, $\overline{V_i}$ is the DJF mean 850 hpa meridional wind over the region of 20–40 $^\circ$N and

110–135 $^\circ$E for year $i$, $\bar{\bar{V}}$ is the mean of $\overline{V_i}$ (Yan et al., 2009).

[4]$\text{EAWMI}\_U_i = \overline{U_{1i}} - \overline{U_{2i}}$, $\overline{U_{1i}}$ is the DJF mean 300 hpa zonal wind over the region of 27.5–37.5 $^\circ$N

and 110–170 $^\circ$E for year $i$, $\overline{U_{2i}}$ is the DJF mean 300 hpa zonal wind over the region of

50–60 $^\circ$N and 80–140 $^\circ$E for year $i$ (Jhun et al., 2004).

[5]$\text{EAWMI}\_P_{1i} = \overline{P_{1i}} - \overline{P_{2i}}$, $\overline{P_{1i}}$ is the DJF mean sea level pressure over the region of 30–55 $^\circ$N and

110–130 $^\circ$E for year $i$, $\overline{P_{2i}}$ is the DJF mean sea level pressure over the region of 20–40 $^\circ$

N and 150–180 $^\circ$E for year $i$ (Yan et al., 2009).

[6]$\text{EAWMI}\_P_{2i} = \overline{P_{1i}}$, $\overline{P_{1i}}$ is the DJF mean sea level pressure over the region of 40–60 $^\circ$N and 80–120 $^\circ$

E for year $i$ (Yan et al., 2009).

**Table 2**. Simulated JJA (DJF) mean surface BC concentrations ($\mu$g m$^{-3}$) in the five

weakest and five strongest EASM (EAWM) years during 1986–2006. Results are

from simulations VMETG4 and VMET averaged over northern China (NC, 110–125 $^\circ$

E, 28–45 $^\circ$N), southern China (SC, 110–125 $^\circ$E, 20–27 $^\circ$N), and eastern China (EC,

110–125 $^\circ$E, 20–45 $^\circ$N).

| Month Region | | Surface Concentrations of BC ($\mu$g m$^{-3}$) | | | | | | | | | |
|---|---|---|---|---|---|---|---|---|---|---|---|
| | | GEOS-4 | | | | | MERRA | | | | |
| | | Weak | Strong | Diff.[a] | Mean[b] | Std.[c] | Weak | Strong | Diff. | Mean | Std. |
| JJA | SC | 0.24 | 0.27 | –0.03 (–11%) | 0.26 | 0.02 | 0.37 | 0.41 | –0.04 (–10%) | 0.39 | 0.02 |
| | NC | 0.94 | 0.85 | 0.09 (11%) | 0.89 | 0.05 | 1.30 | 1.26 | 0.04 (3%) | 1.27 | 0.03 |
| | EC | 0.72 | 0.67 | 0.05 (9%) | 0.70 | 0.03 | 1.02 | 1.00 | 0.02 (2%) | 1.00 | 0.02 |
| DJF | SC | 0.90 | 0.80 | 0.10 (12%) | 0.85 | 0.06 | 1.14 | 1.10 | 0.04 (3%) | 1.12 | 0.04 |
| | NC | 1.76 | 1.63 | 0.13 (8%) | 1.68 | 0.08 | 2.76 | 2.62 | 0.14 (5%) | 2.68 | 0.10 |
| | EC | 1.37 | 1.50 | 0.12 (9%) | 1.43 | 0.07 | 2.26 | 2.15 | 0.11 (5%) | 2.20 | 0.07 |

7 [a]The difference is (Weakest–Strongest) and the relative difference in percentage is in parentheses.

8 [b,c] The mean and the standard deviation of simulated JJA (DJF) mean surface BC concentrations for 1986–2006.

**Table 3**. The composite analyses of JJA (DJF) horizontal fluxes of BC (kg s$^{-1}$) for two selected boxes (northern China (110–125 ° E, 28–45 ° N) and southern China (110–125 °E, 20–27 °N), from the surface to 10 km) based on simulations VMETG4 and VMET. The values are averages over the five weakest and five strongest EASM (EAWM) years during 1986–2006. For horizontal fluxes, positive values indicate eastward or northward transport and negative values indicate westward or southward transport. Also shown are the corresponding wet deposition of BC (kg s$^{-1}$) for the two selected boxes.

| Boundary | GEOS-4 | | | MERRA | | |
|---|---|---|---|---|---|---|
| | Weakest | Strongest | Difference[a] | Weakest | Strongest | Difference[a] |
| JJA, northern China (110–125 °E, 28–45 °N) | | | | | | |
| South | +2.24 | +0.97 | +1.27 | +1.93 | +0.92 | +1.01 |
| North | +3.44 | +4.06 | –0.62 | +3.90 | +4.57 | –0.67 |
| West | +6.60 | +4.20 | +2.40 | +8.72 | +7.51 | +1.21 |
| East | +12.48 | +9.20 | +3.28 | +3.60 | +2.31 | +1.29 |
| Net | inflow 1.01 | | | inflow 1.60 | | |
| Deposition | 14.06 | 13.35 | 0.70 | 13.26 | 11.76 | 1.50 |
| JJA, southern China (110–125 °E, 20–27 °N) | | | | | | |
| South | +0.62 | +0.70 | –0.08 | +0.61 | +0.60 | +0.01 |
| North | +1.79 | +0.88 | +0.91 | +1.67 | +0.95 | +0.72 |
| West | +0.94 | +0.13 | +0.81 | +0.47 | +0.12 | +0.35 |
| East | +0.33 | +0.42 | –0.09 | +0.18 | +0.27 | –0.09 |
| Net | outflow 0.09 | | | outflow 0.27 | | |
| Deposition | 2.46 | 3.02 | –0.56 | 2.26 | 2.84 | –0.58 |

DJF, northern China (110–125 °E, 28–45 °N)

| | | | | | | |
|---|---|---|---|---|---|---|
| South | −6.35 | −8.24 | +1.89 | −4.51 | −5.96 | +1.45 |
| North | −0.37 | −0.71 | +0.34 | +0.64 | −0.28 | +0.92 |
| West | +11.60 | +11.41 | +0.19 | +12.01 | +12.90 | −0.89 |
| East | +22.77 | +21.67 | +1.10 | +23.55 | +24.53 | −0.98 |
| Net | inflow 0.64 | | | inflow 0.62 | | |
| Deposition | 9.48 | 9.24 | 0.24 | 9.17 | 8.75 | 0.42 |

DJF, southern China (110–125 °E, 20–27 °N)

| | | | | | | |
|---|---|---|---|---|---|---|
| South | −3.09 | −3.61 | +0.52 | −2.77 | −3.47 | +0.70 |
| North | −5.23 | −6.68 | +1.45 | −4.40 | −5.59 | +1.19 |
| West | +1.03 | −0.64 | +1.67 | +1.24 | +0.25 | +0.99 |
| East | +2.68 | +2.13 | +0.55 | +0.98 | +0.88 | +0.10 |
| Net | inflow 0.19 | | | inflow 0.40 | | |
| Deposition | 4.78 | 4.52 | 0.26 | 4.79 | 4.51 | 0.28 |

1    [a]The difference is (Weakest–Strongest).

**Table 4**. Simulated JJA (DJF) mean all-sky direct radiative forcing (DRF) of BC (W

m$^{-2}$) at the top of the atmosphere (TOA) in the five weakest and five strongest EASM

(EAWM) years during 1986–2006. Results are from simulation VMET averaged over

eastern China (110–125 °E, 20–45 °N), northern China (110–125 °E, 28–45 °N), the

northern China Plain (110–125 °E, 37–45 °N), the central China Plain (110–125 °E,

28–36 °N), and southern China (110–125 °E, 20–27 °N).

| Month | Region | TOA DRF of BC, MERRA (W m$^{-2}$) | | |
|-------|--------|------|--------|------------|
| | | Weak | Strong | Difference[a] |
| JJA | southern China | 0.34 | 0.40 | −0.06 (14%) |
| | northern China | 1.41 | 1.38 | 0.04 (3%) |
| | eastern China | 1.08 | 1.07 | 0.01 (1%) |
| DJF | southern China | 1.04 | 1.07 | −0.03 (3%) |
| | northern China | 1.65 | 1.62 | 0.03 (2%) |
| | central China Plain | 2.11 | 2.14 | −0.03 (1%) |
| | northern China Plain | 1.08 | 0.97 | 0.11 (11%) |
| | eastern China | 1.46 | 1.45 | 0.01 (1%) |

8 [a]The difference is (Weakest–Strongest) and the relative difference in percentage is in parentheses.

**Table 5**. The composite analyses of DJF horizontal and vertical fluxes of BC (kg s$^{-1}$)

for two selected boxes (the central China Plain (110–125 °E, 27–36 °N) and southern

China (110–125 °E, 20–27 °N), from 1 to 6 km) based on simulation VMET. The

values are averages over the five weakest and five strongest EAWM years during

1986–2006. For fluxes, positive values indicate eastward, northward, or upward

transport and negative values indicate westward, southward, or downward transport.

| Boundary | Weakest | Strongest | Difference[a] | Net |
|---|---|---|---|---|
| DJF, central China Plain (110–125 °E, 28–36 °N) | | | | |
| South | +1.29 | +0.98 | +0.31 | Inflow 0.01 |
| North | +0.53 | +0.07 | +0.46 | |
| West | +7.84 | +8.89 | −1.05 | |
| East | +7.39 | +8.61 | −1.21 | |
| Upper | +0.99 | +1.24 | −0.25 | outflow 0.11 |
| Bottom | +5.22 | +5.56 | −0.34 | |
| DJF, southern China (110–125 °E, 20–27 °N) | | | | |
| South | −0.08 | −0.20 | +0.12 | inflow 0.03 |
| North | +0.91 | −0.67 | +0.24 | |
| West | +4.40 | +4.37 | +0.03 | |
| East | +1.70 | +1.82 | −0.12 | |
| Upper | +0.09 | +0.06 | +0.03 | outflow 0.07 |
| Bottom | +1.12 | +1.16 | −0.04 | |

7 [a]The difference is (Weakest–Strongest)

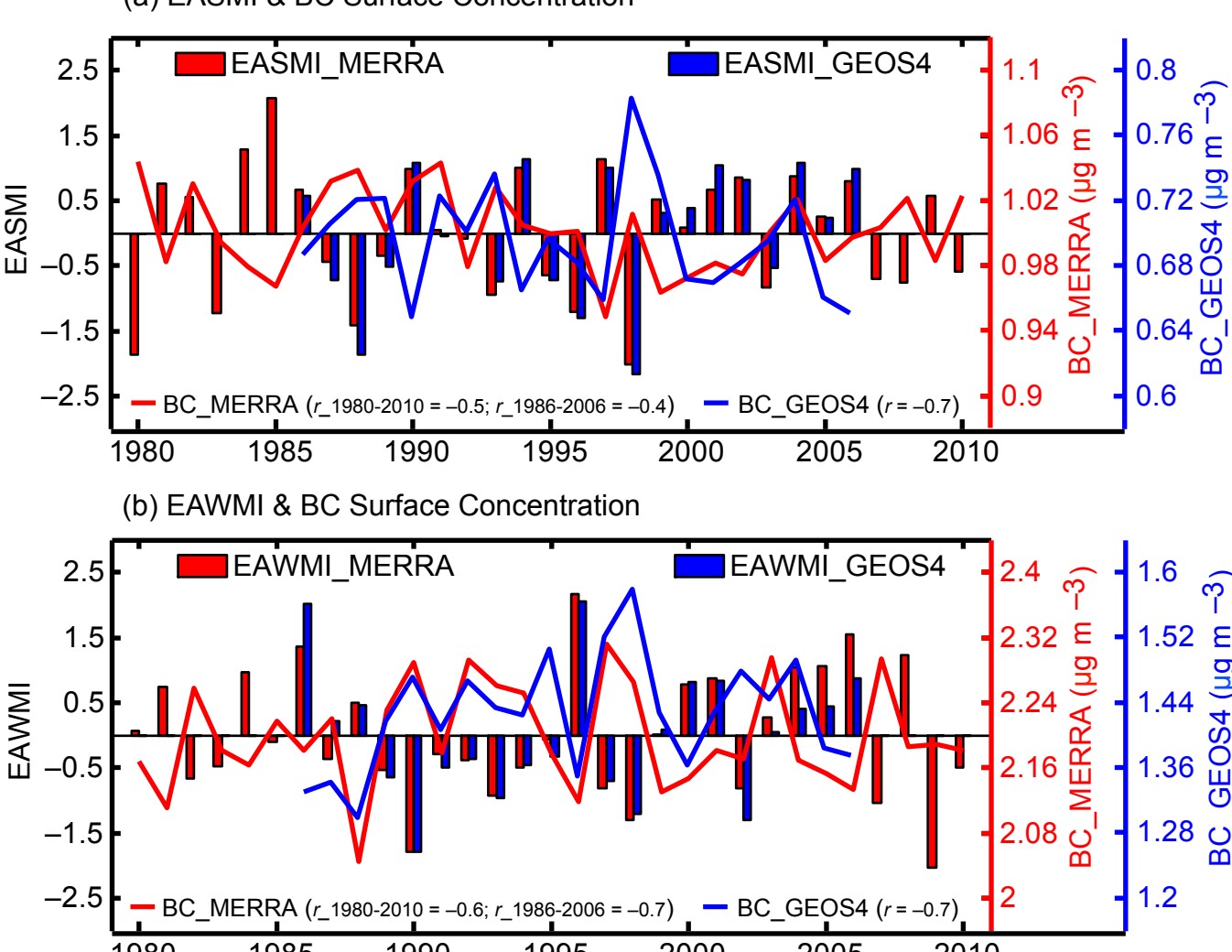

**Fig. 1. (a)** Normalized East Asian summer monsoon Index (EASMI, bars, left y axis) and the simulated June-July-August (JJA) mean surface BC concentrations (lines, right y axis, $\mu g\ m^{-3}$) averaged over eastern China (20–45° N, 110–125° E) from model simulation VMET (red line) for 1980–2010 and from VMETG4 (blue line) for 1986–2006. EASMI are calculated based on MERRA (red bars) and GEOS-4 (blue bars) assimilated meteorological data following Li and Zeng (2002). **(b)** Same as **(a)**, but for normalized East Asian winter monsoon Index (EAWMI) and the simulated December-January-February (DJF) mean surface BC concentrations. EAWMIs are calculated following Wu and Wang (2002).

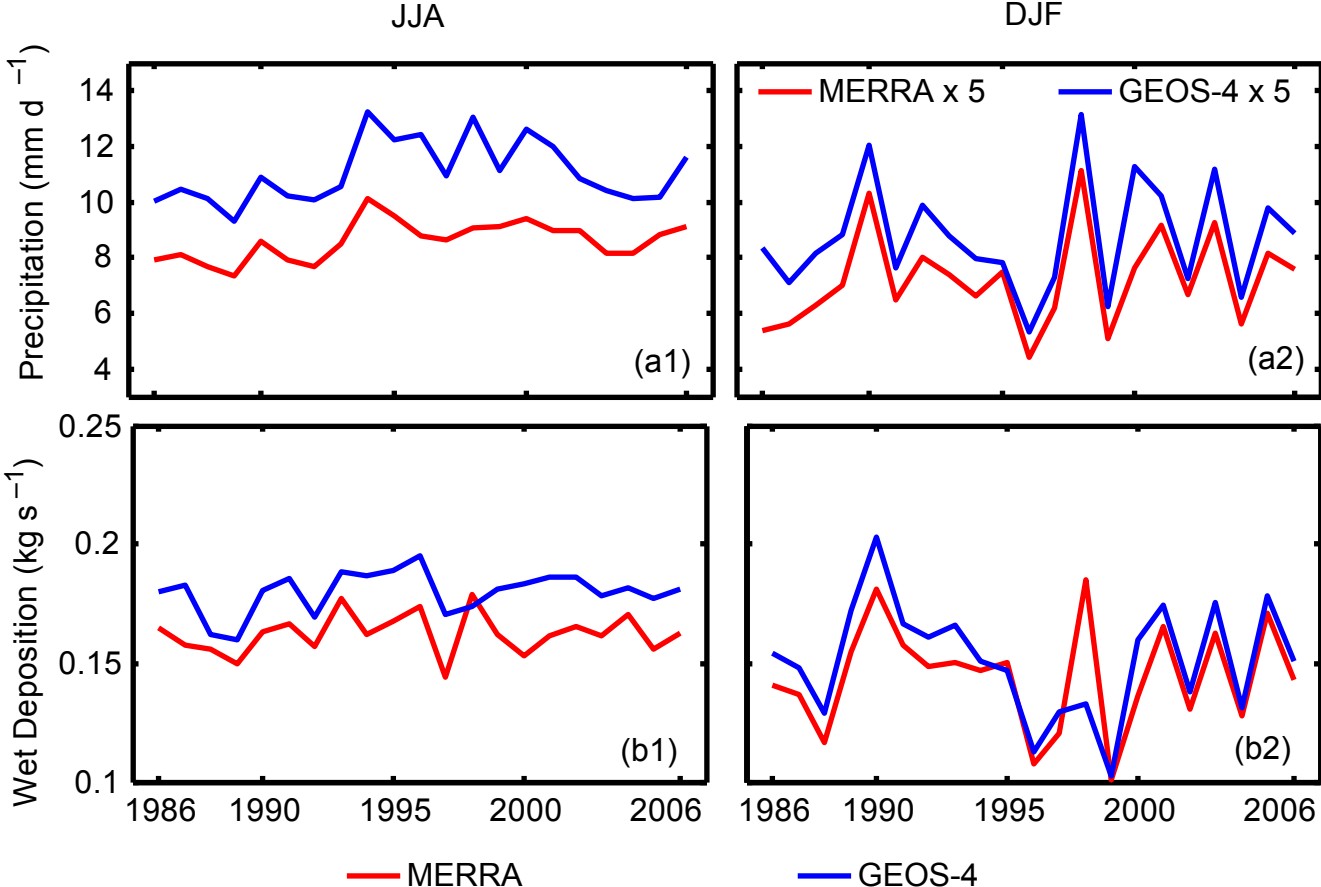

**Fig. 2. (a)** JJA and DJF mean precipitation (mm d$^{-1}$) averaged over eastern China for 1986–2006 from GEOS-4 (blue lines) and MERRA (red lines) meteorological data. DJF mean precipitation is multiplied by 5 in **(a2)**. **(b)** Same as **(a)**, but for wet deposition (kg s$^{-1}$).

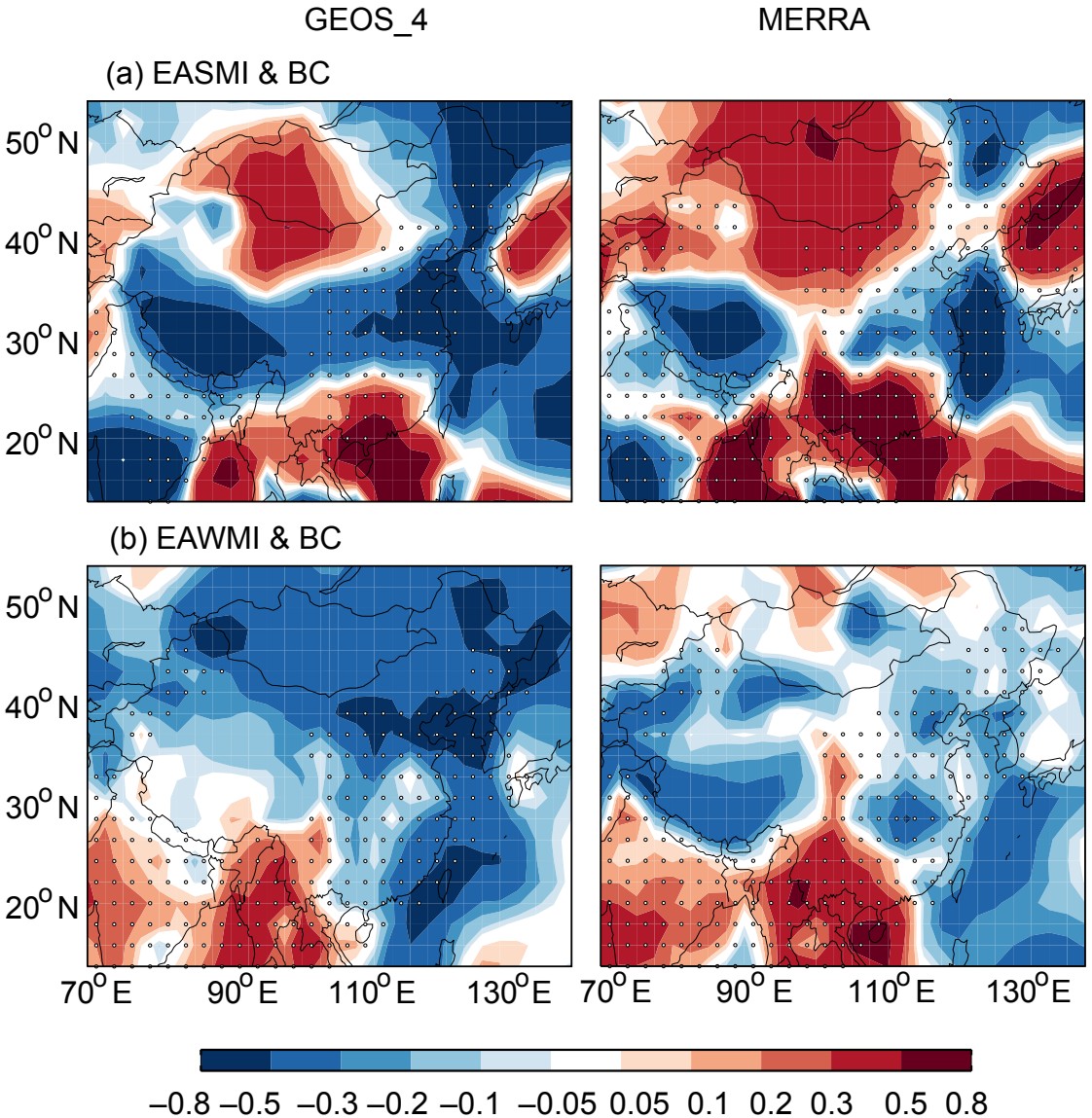

**Figure 3.** **(a)** Correlation coefficients between EASMI and JJA mean surface BC concentrations during 1986–2006. **(b)** Correlation coefficients between EAWMI and DJF mean surface BC concentrations during 1986–2006. Simulated BC concentrations are from model simulations VMETG4 (left) and VMET (right), and monsoon indexes are calculated based on GEOS-4 (left) and MERRA (right) assimilated meteorological data. The dotted areas indicate statistical significance with 95% confidence from a two-tailed Student's t test.

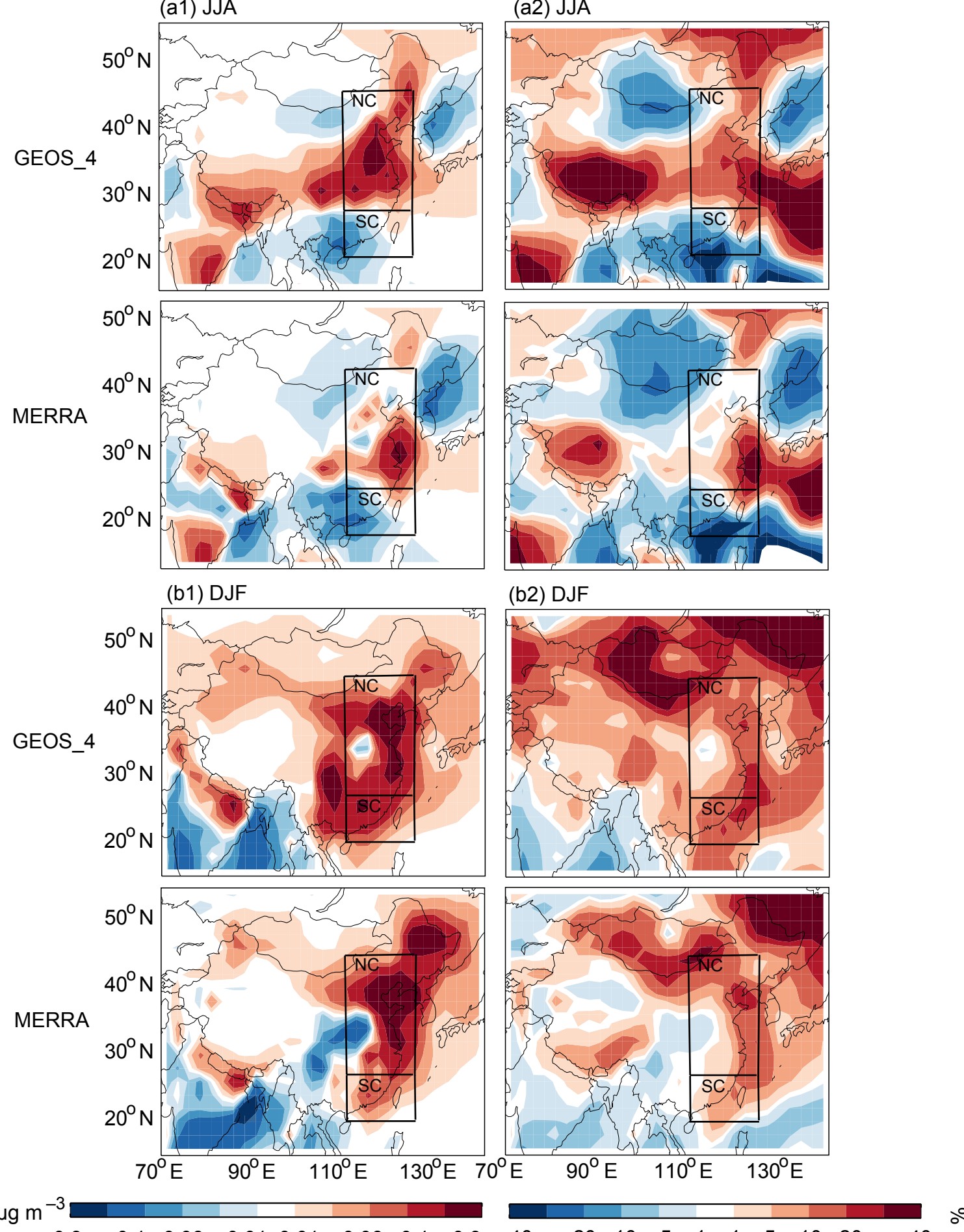

**Fig. 4**

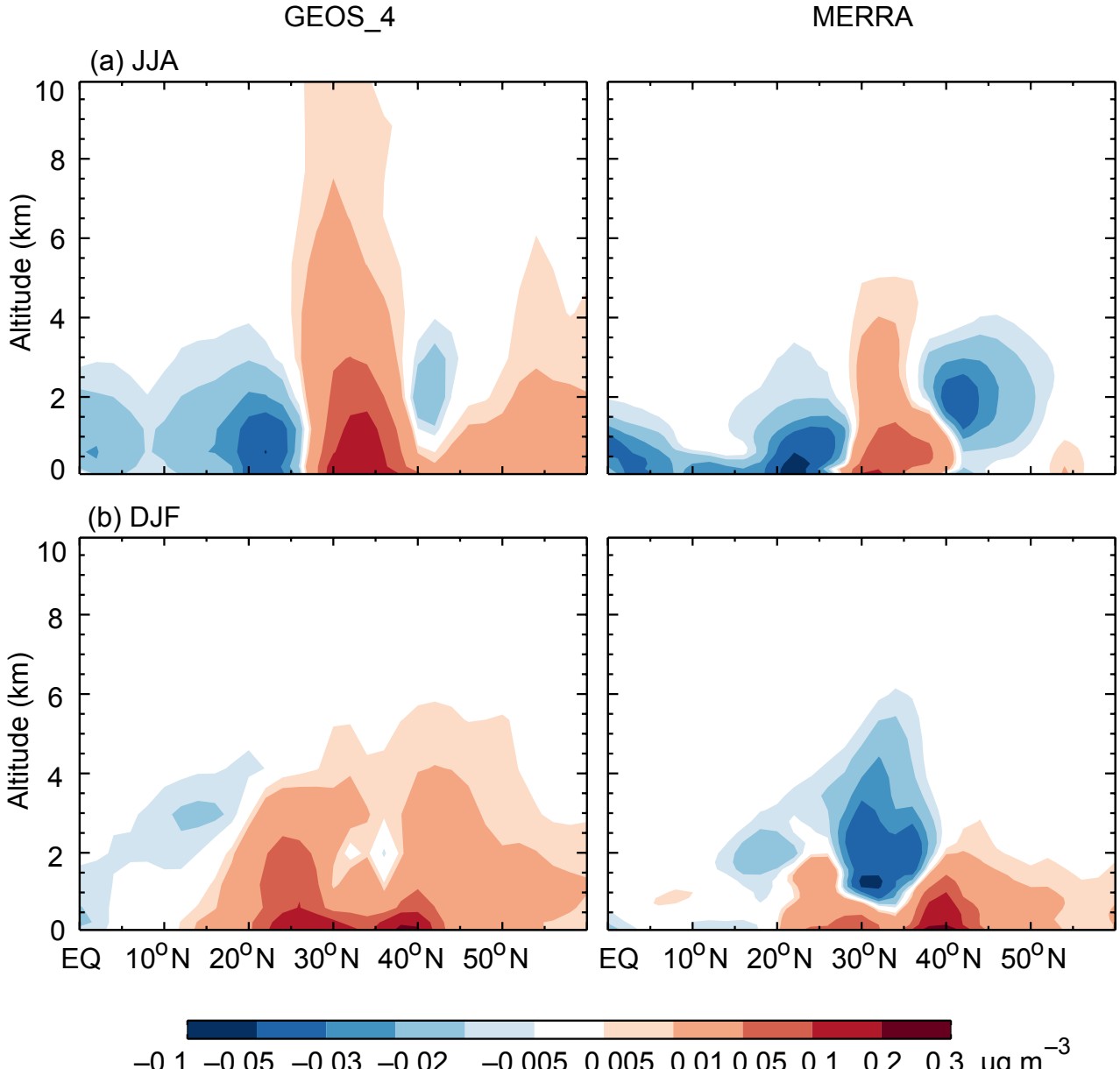

**Fig. 5. (a)** Height-latitude cross section of differences in simulated JJA mean BC concentrations ($\mu$g m$^{-3}$) between the five weakest and five strongest EASM years during 1986–2006. Plots are averaged over longitude range of 110–125° E from model simulations VMETG4 (left) and VMET (right). **(b)** Same as **(a)**, but for differences in DJF between five weakest and five strongest EAWM years.

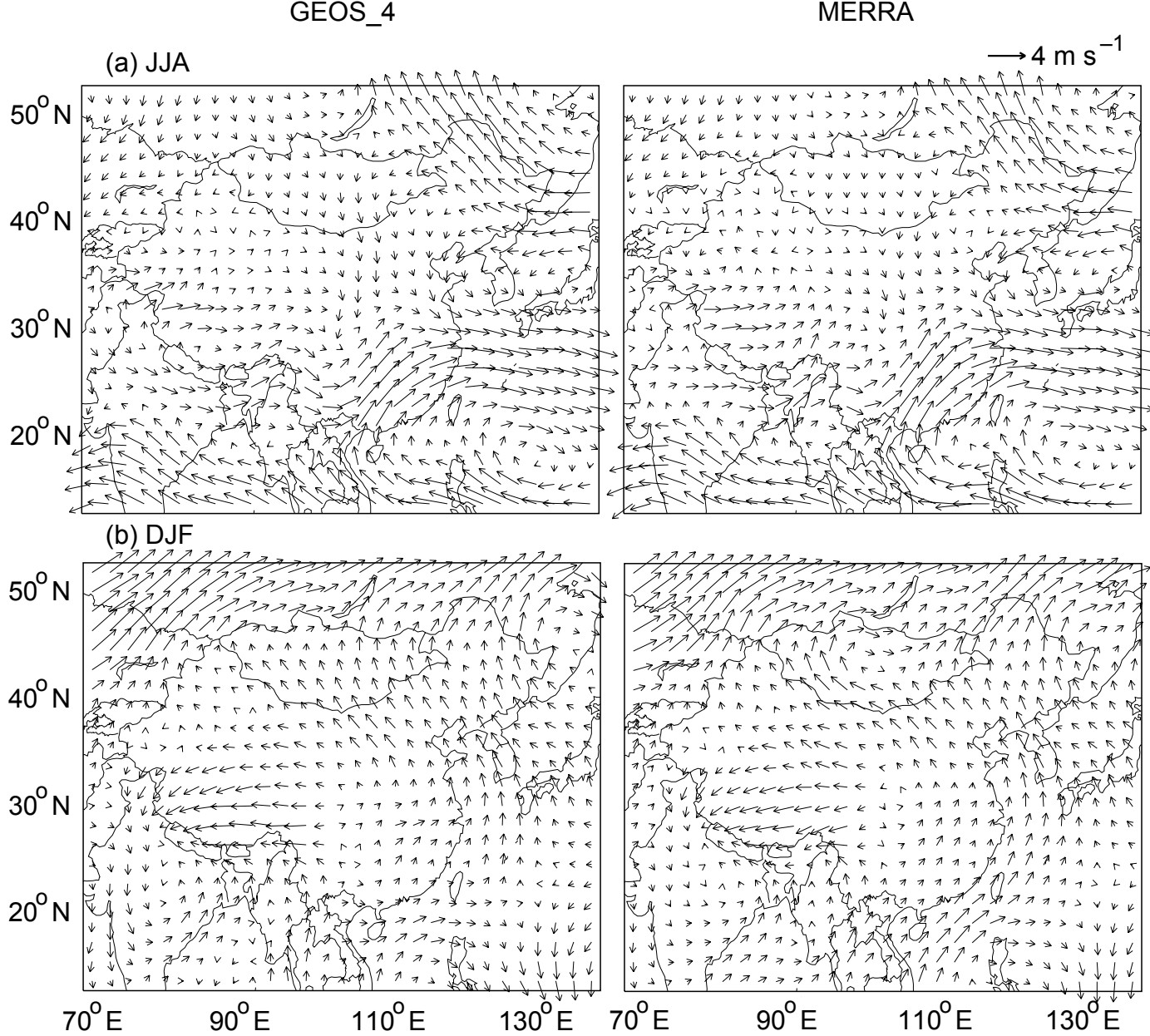

**Fig. 6. (a)** Differences in JJA 850 hPa wind (vector, m s$^{-1}$) between the five weakest and five strongest EASM years during 1986–2006 from GEOS-4 (left) and MERRA (right) data. **(b)** Same as **(a)**, but for differences in DJF wind between five weakest and five strongest EAWM years.

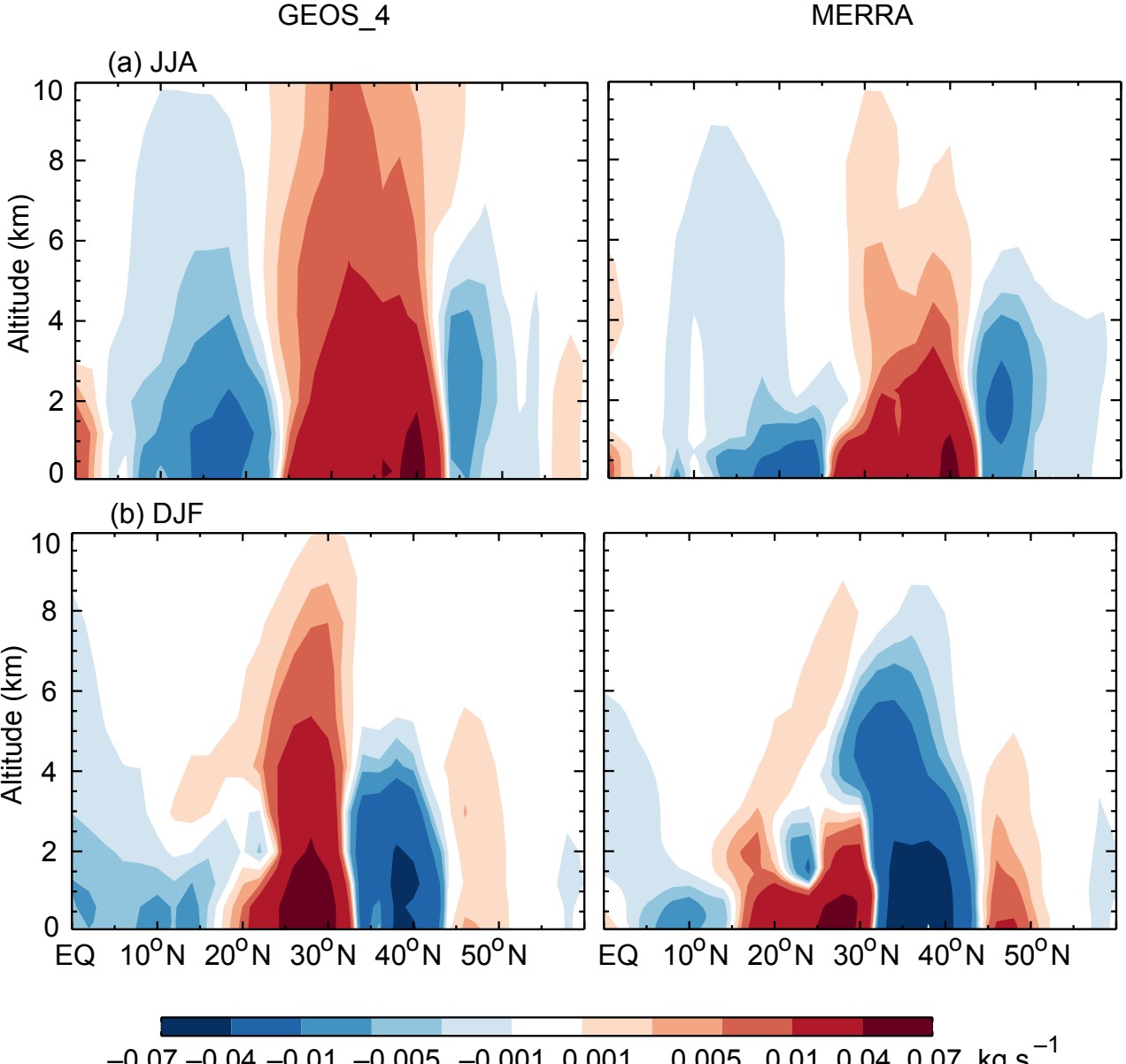

**Fig. 7. (a)** Differences in simulated upward mass flux of JJA BC (kg s$^{-1}$) between the five weakest and five strongest EASM years during 1986–2006. Plots are averaged over longitude range of 110–125° E from model simulations VMETG4 (left) and VMET (right). **(b)** Same as **(a)**, but for differences in DJF between five weakest and five strongest EAWM years.

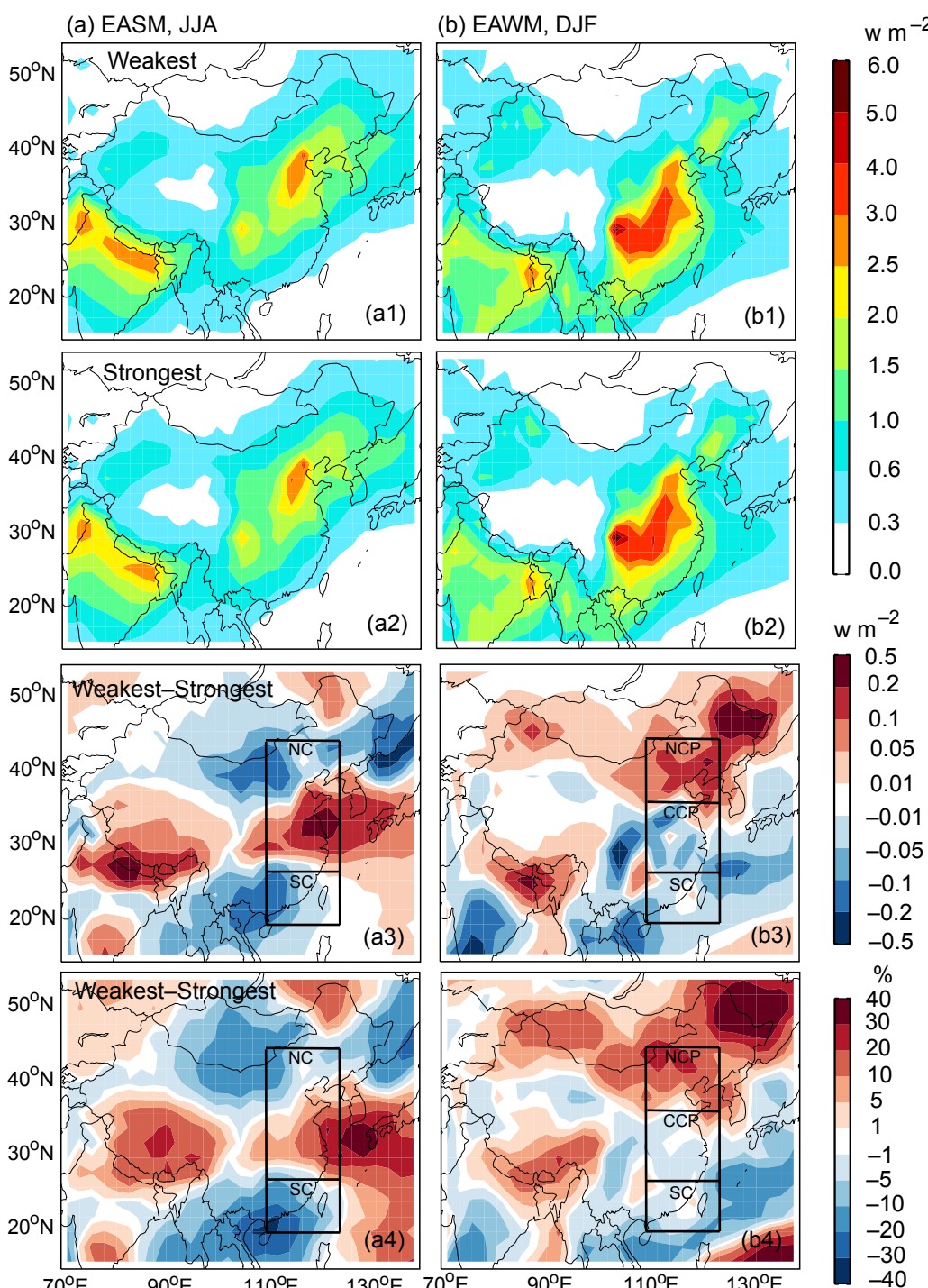

**Fig. 8. (a)** Simulated JJA mean all-sky direct radiative forcing (DRF) of BC (W m$^{-2}$) at the top of the atmosphere (TOA) in the **(a1)** five weakest and **(a2)** five strongest EASM years during 1986–2006 from model simulation VMET. Also shown are the **(a3)** absolute (W m$^{-2}$) and **(a4)** percentage (%) differences between the five weakest and five strongest EASM years. **(b)** Same as **(a)**, but for simulated DJF mean all-sky TOA DRF of BC in the five weakest and five strongest EAWM years. The enclosed areas are defined as northern China (NC, 110–125° E, 28–45° N), the northern China Plain (NCP, 110–125° E, 36–45° N), the central China Plain (CCP, 110–125° E, 28–36° N), and southern China (SC, 110–125° E, 20–27° N).

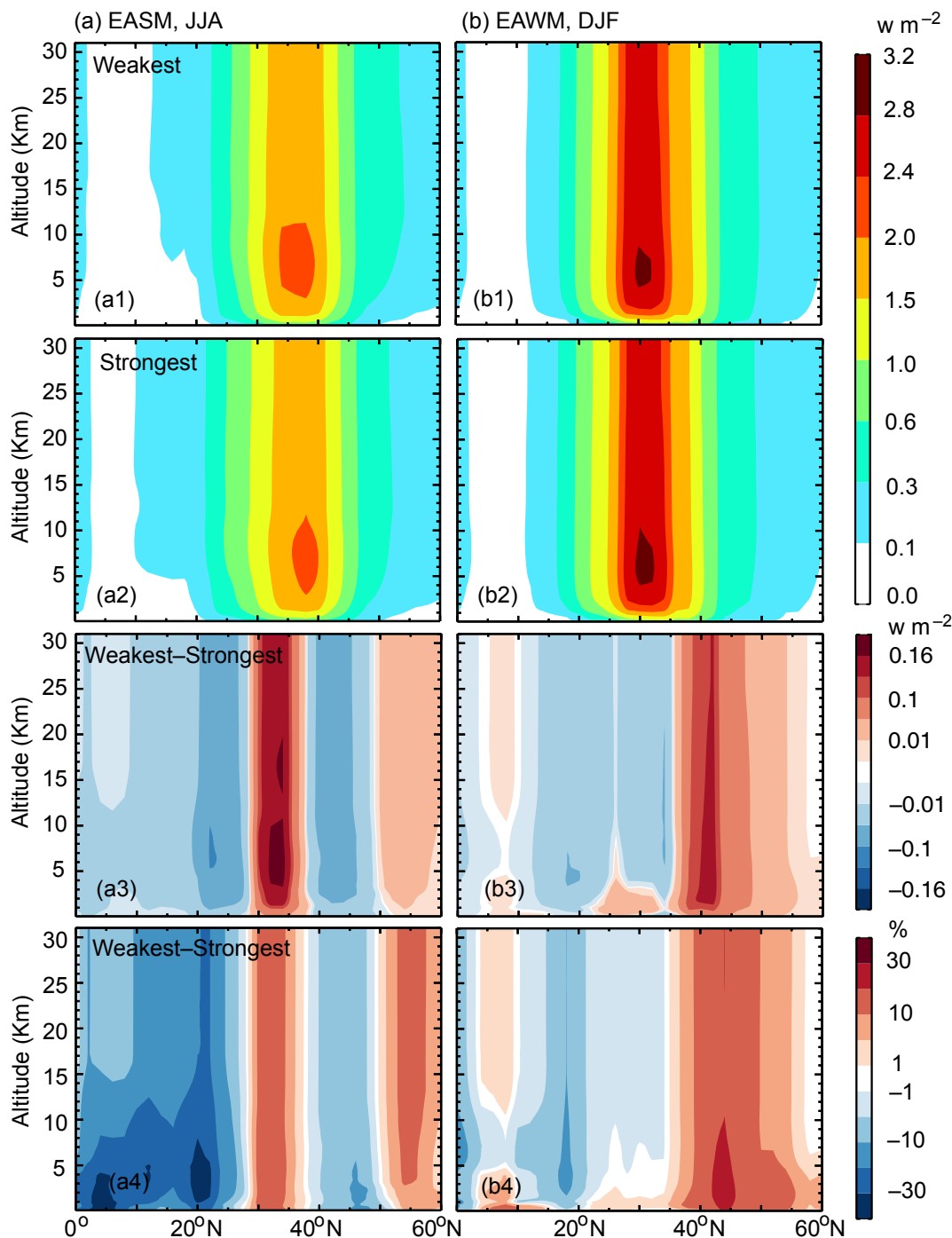

**Fig. 9. (a)** Height-latitude cross sections of simulated JJA mean all-sky DRF of BC (W m$^{-2}$) in the **(a1)** five weakest and **(a2)** five strongest EASM years during 1986–2006. Also shown are the **(a3)** absolute (W m$^{-2}$) and **(a4)** percentage (%) differences between the five weakest and five strongest EASM years. Plots are averaged over longitude range of 110–125° E from model simulation VMET. **(b)** Same as **(a)**, but for simulated DJF mean all-sky DRF of BC in the five weakest and five strongest EAWM years.

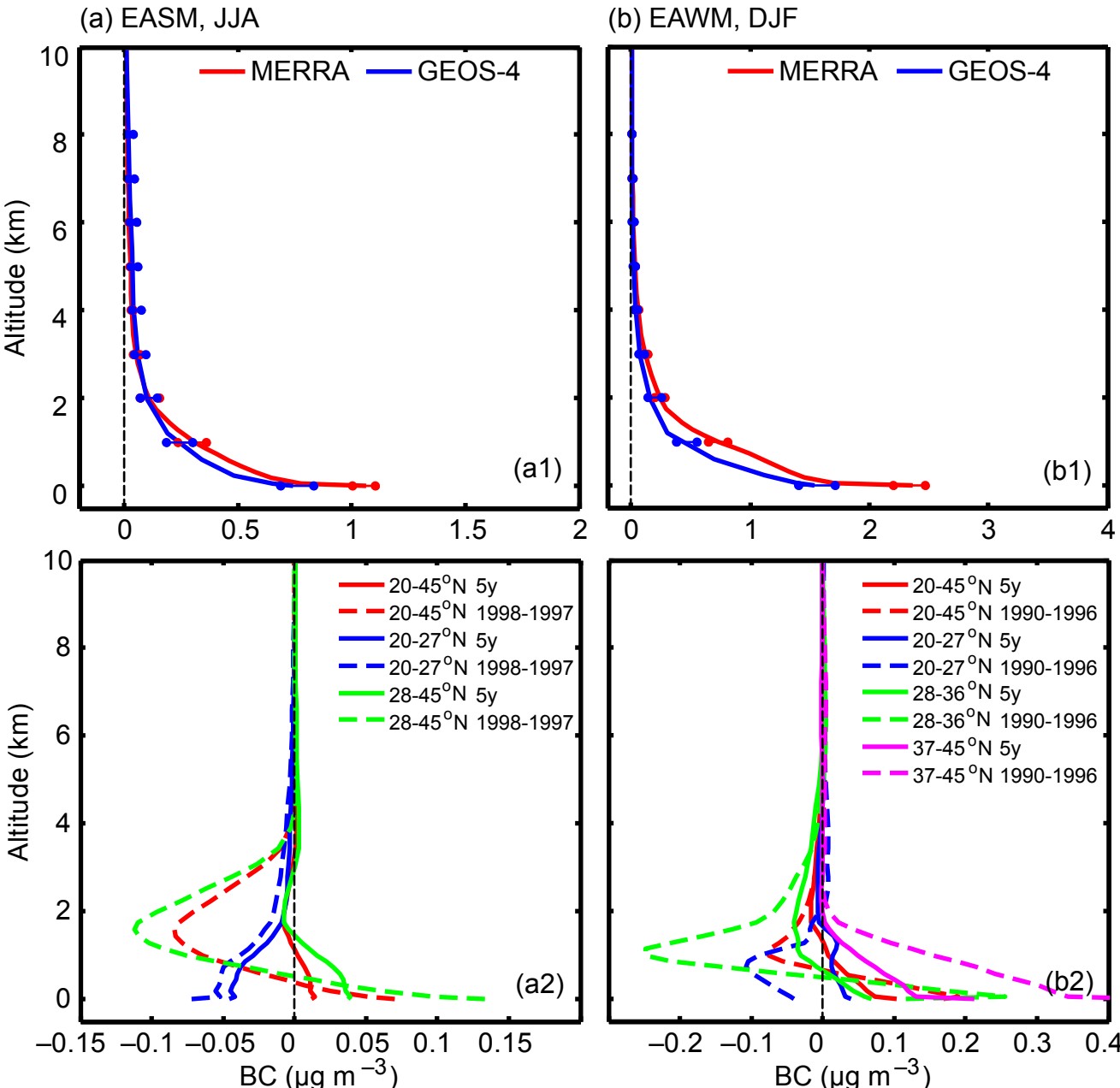

**Fig. 10. (a1)** Simulated vertical profiles of JJA BC mass concentrations ($\mu g\ m^{-3}$) averaged over 1986–2006. The error bars represent the minimum and maximum values of BC. Results are averages over eastern China from model simulations VMETG4 (blue) and VMET (red). **(a2)** Differences in simulated vertical profiles of JJA BC mass concentrations ($\mu g\ m^{-3}$) between the five weakest and five strongest EAM years (solid lines) during 1986–2006, and between the weakest and strongest EASM years (1998–1997, dotted lines). Results are averages over eastern China, northern China, and southern China from model simulations VMET. **(b1)** Same as **(a1),** but for simulated DJF BC mass concentrations. **(b2)** Same as **(a2),** but for differences in DJF between the five weakest and five strongest EAWM years and between the weakest and strongest EAWM years (1990–1996). Results are averages over eastern China, northern China Plain, the central China Plain, and southern China.

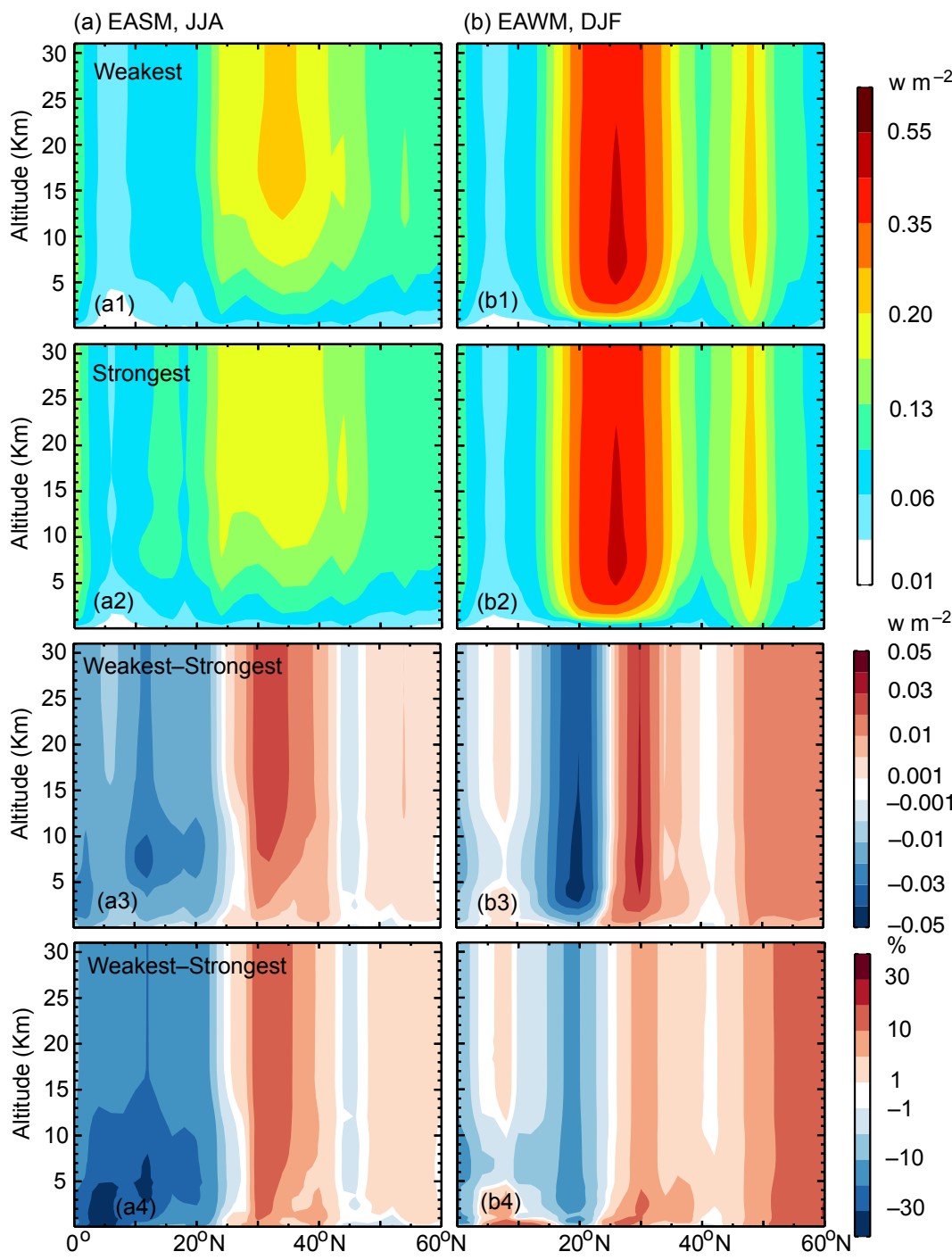

**Figure 11.** Same as **Figure 9,** but for the contributions from non-China emissions to simulated all sky DRF of BC .

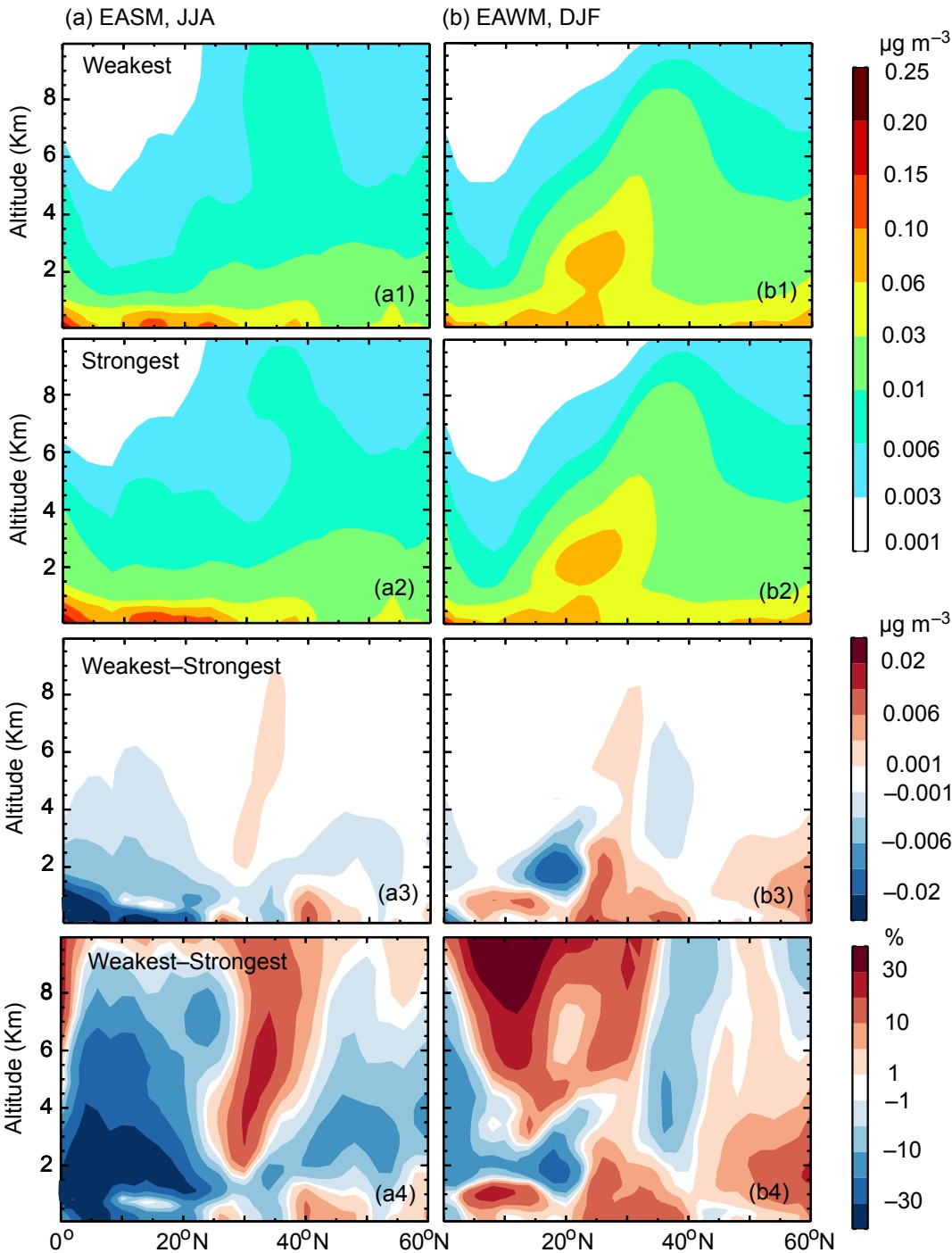

**Fig. 12. (a)** Height-latitude cross sections of contributions of non-China emissions to simulated JJA mean BC concentrations (μg m$^{-3}$) in the **(a1)** five weakest and **(a2)** five strongest EASM years during 1986–2006. Also shown are the **(a3)** absolute (μg m$^{-3}$) and **(a4)** percentage (%) differences between the five weakest and five strongest EASM years. Plots are averaged over longitude range of 110–125° E from model simulation VMET. **(b)** Same as **(a)**, but for simulated DJF mean BC concentrations in the five weakest and five strongest EAWM years.

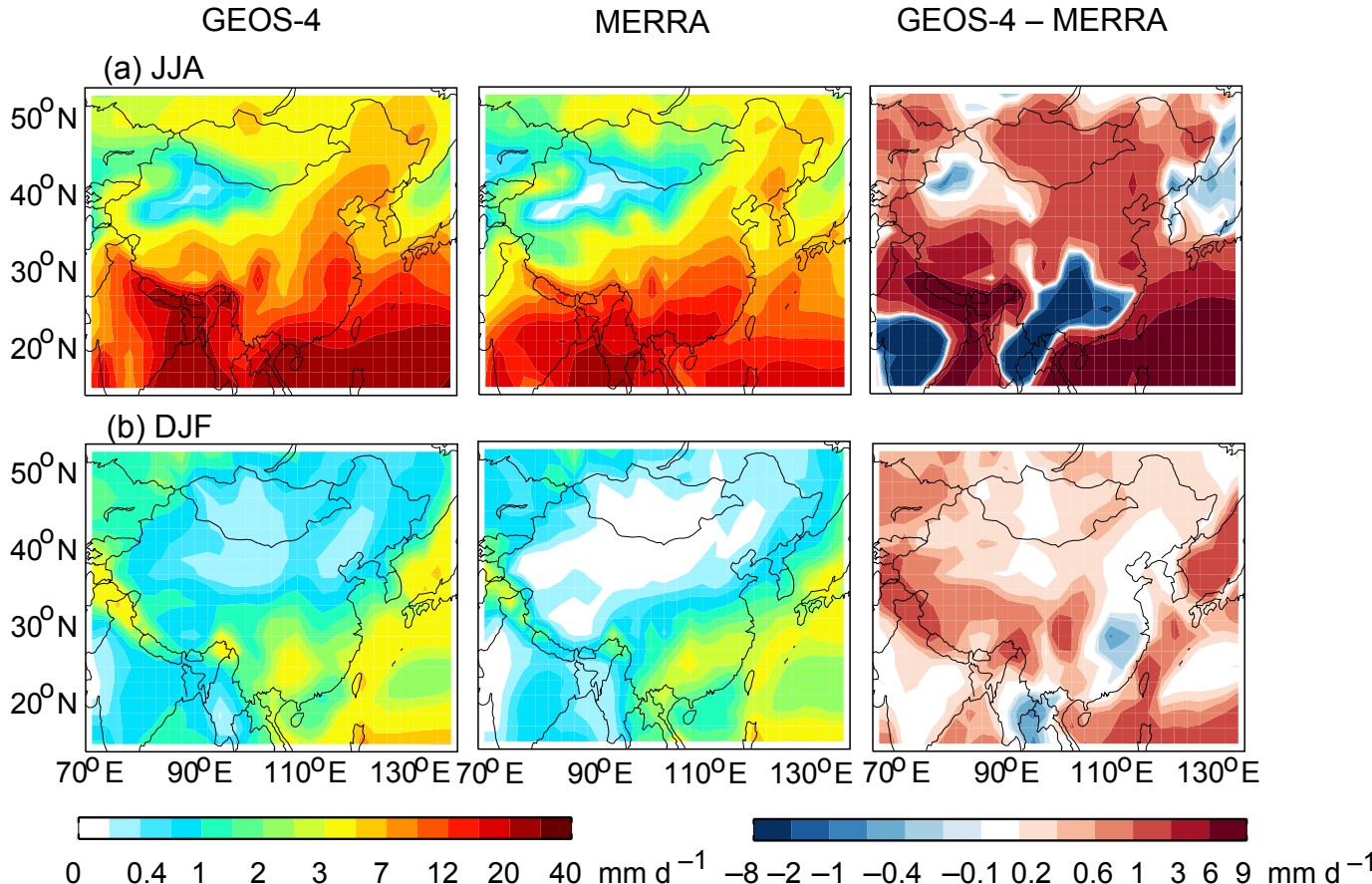

**Fig. 1S .** JJA and DJF mean precipitation (mm d$^{-1}$) averaged for 1986–2006 from GEOS-4 **(a)** and MERRA **(b)** meteorological data. Also shown are the differences between GEOS-4 and MERRA data **(c)**.

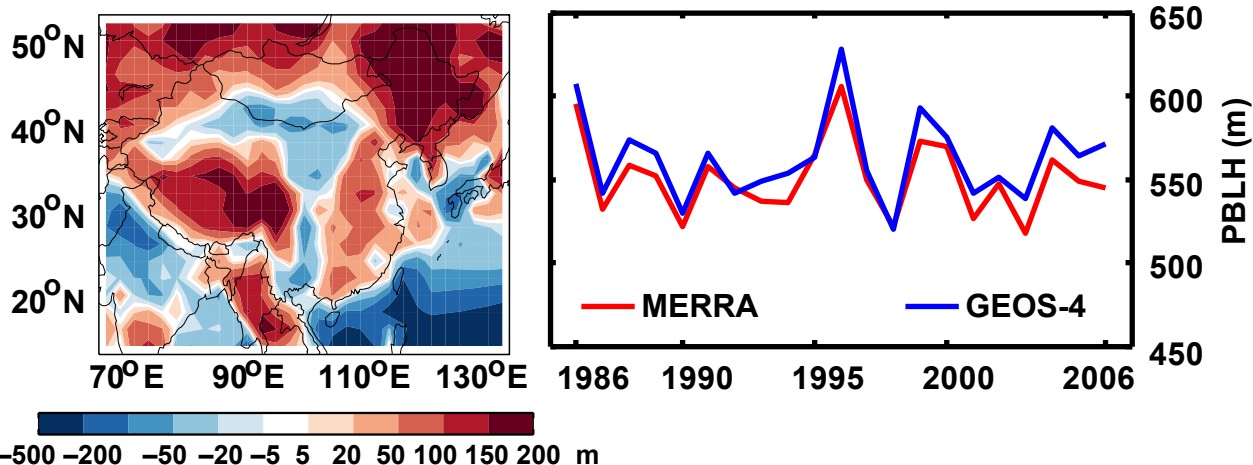

**Fig. 2S. (left)** Differences in DJF mean planetary boundary layer height (PBLH, m) averaged for 1986–2006 between GEOS-4 and MERRA. **(right)** DJF mean PBLH averaged over eastern China for 1986–2006 from GEOS-4 (blue line) and MERRA (red line).