# Peer review of "Impacts of East Asian Summer and Winter Monsoon on"

_Atmospheric Chemistry and Physics, 2016_

## Referee Comment (RC1) · Anonymous Referee #2 · 20 Jun 2016

In this work authors attempted to study the impacts of the interannual variation of Eastern Asian summer and winter monsoon on variations of black carbon (BC) mass concentrations and direct radiative forcing (DRF) in Eastern China during 1986-2006. Overall this paper is quite lengthy and reads more like a technical report. The results presented in the paper solely rely on model simulations lack of any observational evidences or cross-validation with previous modeling studies of BC. Some issues with respect to the method descriptions sound vague. The clarification of these issues is critical to understand comprehensive results presented in this study. I recommend the major revision of the paper before the possible acceptance of ACP by addressing my

following comments.

Major comments:

1. The methodology used in this study simply followed previous studies [Zhu et al., 2012; Yang et al. 2014]. That's fine. The results of BC are not surprising to me at all since BC is one of important fine aerosol types (i.e., PM2.5) discussed in Zhu et al. [2012]. It might be more interesting to emphasize the change of special characteristics of BC (e.g., whether or how the change of cloud layer between weakest and strongest Eastern Asian monsoon impacts on the BC absorption and DRF).

2. The results presented in the paper solely rely on model simulations lack of any observational evidences or cross-validation with previous modeling studies of BC. This makes me wonder how the modeled BC mass concentrations in this work compare with historical observations available in Eastern China, especially during JJA and DJF (i.e., the seasons authors focus on in this work).

3. Authors presented major results based on the difference between weakest and strongest Eastern Asia summer monsoon in Section 3 (covering Fig. 1a, Fig. 2a1, 2b1, Fig. 3a, . . .) and then from the difference in winter monsoon in Section 4 (covering Fig. 1b, Fig. 2a2,2b2, Fig. 3b, . . .). However, no discussions (linked to changes in winds or circulation patterns, etc) were made on the difference between summer and winter monsoon, which makes two sections sound like separate stories.

4. Majority results in this work (i.e., Fig. 4-12, Table 2-5) highly reply on the difference between weakest and strongest Eastern Asia summer monsoon (in Section 3). The selection of five weakest and strongest years in this work is slightly different with previous studies [Zhu et al., 2012; Yang et al. 2014] that used the same GEOS-4 met fields of 1986-2006 without any explanations. Please explain why authors choose different monsoon years as adopted in Zhu et al. 2012 and Yang et al. 2014.

5. On Page 5 Line 26, should 1980-2010 be 1986-2006, which overlaps the period

between GEOS-4 and MERRA?

6. On Page 8 Line 5-6, authors mentioned numerous studies have shown that the intensity of EAWM. . . . . .but they only cited one reference of Yan et al. (2009). It sounds contradictory.

7. In Section 3 and 4, authors enclosed values in the parenthesis but did not describe how they calculate these values, for instance the range of percentages on Page 8 Lines 22-23. Please add the clarification.

8. On Page 10 Lines 19-20, what is the cause of the different pattern of BC concentration between GEOS-4 and MERRA shown in Figure 5a?

9. On Page 11 Lines 2-3, how does the convergence cause the increase in BC concentration and anticyclone wind pattern cause the decrease in BC concentration?

10. On Page 11, Lines 8-10, I understand the convergence accompanied with the descending air prevents surface BC to the upper troposphere, causing the increase in surface BC. But I don't understand why the upward mass flux of BC also increases under the condition of convergence. Could you explain it?

11. On Page 11 Line 13, please describe the method you calculate horizontal mass fluxes at the four lateral boundaries in details. Clearly, the net effect does not equal to the fluxes summed with values from four lateral boundaries. How do you calculate the net effect of horizontal mass fluxes over the specific region?

12. On Page 11 Lines 23-25, why is there larger inflow at the east and north boundary and smaller outflow at the south and east boundary?

13. On page 11 Lines 27-29, where do these two numbers (i.e., 0.09 and 0.27 kg/s) come from? Do you average them over the entire domain? Please specify the region your numbers are based on?

14. On Page 13, Lines 4-5, could you clarify what is the direct radiative forcing efficiency of BC? BTW, did you notice that the shift of the center of the highest BC DRF from weakest to strongest? Could you explain what is the cause of the shift?

15. On Page 13, Lines 18-21, please add quantitative metrics to quantify the change of BC DRF in northern and southern China.

16. On Page 13, Lines 26-27, how do you distinguish the DRF of BC between non-China emission and local sources? Did you offline run the radiative transfer model? If yes, please describe it in the section of method.

17. On Page 14, line 22, could you show PBLH in the supplement? Also explain how PBLH changes surface BC concentration.

18. On Page 15, Lines 23-24, what is the cause of the different response of BC concentration to the summer and winter monsoon in southern china?

19. On Page 17, lines 21-24. I cannot tell the lower column burden of tropospheric BC from your Figure 5b. It appears that the BC profile increase at all altitudes. Is it related to the change of clouds?

20. On Page 17, Lines 24-27, why is DRF lower in the weakest monsoon years in southern china even though both BC surface concentration and column burden are higher, compared with the strongest monsoon years?

21. On Page 20, besides simply reporting what you conclude in this work, could you add some discussion about why eastern Asian summer and winter monsoon change BC concentration and DRF in northern and southern China differently? Is this difference important to contribute to the air quality regulation in different regions of China?

22. On Page 5 Lines 17-18, BC is assumed externally mixed with other aerosol species in this model. Could authors discuss the uncertainties of your results based on this assumption? How do results change if BC is partially internally or internally mixed with other aerosol species?

Minor comments:

1. Page 11 Line 12, change "summary" to "summarize".

2. Figure 1, add the description of r31y and r21y.

3. Figure 4, please move the row of a2 above b1 since you discussed a2 ahead of b1 in the context.

4. Figure 10, please label a1, a2, b1, and b2 in Figure.

5. Figure 12, How do you distinguish BC concentration attributed to non-China emissions and local China sources in the model? Please specify in the description of the model.

References:

Yang, Y., Liao, H., and Li, J.: Impacts of the East Asian summer monsoon on interannual variations of summertime surface-layer ozone concentrations over China, Atmos. Chem. Phys., 14, 6867–6880, doi:10.5194/acp-14-6867-2014, 2014.

Zhu, J., Liao, H., and Li, J.: Increases in aerosol concentrations over eastern China due to the decadal-scale weakening of the East Asian summer monsoon, Geophys. Res. Lett., 39, L09809, doi:10.1029/2012GL051428, 2012.

---

## Referee Comment (RC2) · Anonymous Referee #3 · 25 Jun 2016

This study used a chemistry transport model to examine the impact of monsoon variability on black carbon distribution (surface concentration, vertical profile and direct forcing as an integrated measure) over China. Since the emissions are fixed in the model simulation, metrological variability is the dominant sources of pollution variability. The results of this study generally support many empirical analyses that linked observed pollution level with monsoon variability as extensively cited in the paper. I found the analysis is comprehensive and the interpretation of the results is convincing. I recommend publication after the following issues are addressed.

[Figure]

One of my main concerns is regard to the interpretation of summer monsoon impact. It is noted by the authors " differences in   transport of BC due to the changes in atmospheric circulation are a dominant mechanism through which the EASM influences the variations of JJA BC". However, the role of precipitation in setting wet removal is not adequately tested (e.g. excessive rainfall  in strong monsoon year can increase wet removal flux and also contribute to pollution reduction). Of course, I admit that precipitation can also be correlated to circulation and moisture transport, so a deeper question would be How do you separate the effects of circulation (dispersion) vs. rainfall (removal)?

For wintertime, it is reasonable that circulation is the main cause due to lack of rainfall generally. However, in explaining the higher surface concentrations in weaker monsoon years, how do you separate the effects of (1) weaker horizontal transport that leads to higher total column loading of pollution buildup and (2) weaker vertical mixing that tends to put more pollution at the surface? Are these two processes working in the same direction or now? Which is more important in the model?

How do you account for the warming trends in the simulation years? Are the temperature/precipitation trends significant? Are they leading to trends in pollutions?

Minor points:

Page 1. Line 9. How do these affect "influence the variations of emissions"? Line 19. "convention" -> "convection"

Page 2. Line 24. This is just repeating the reference in page 1?

Page 4. Line 7. "matters" -> "matter"

Page 5. Line 5. What's a more appropriate reference here than Ramanathan and Carmichael (2008) is Ramanathan and Xu (2010). Line 11. It is odd to compare 1980-2010 forcing over Asia with 1850-2005 forcing globally.

Line 14. "Changes in monsoon"? But you did not provide references that monsoon was

indeed changing over EA. Line 22. If there is any statistical analysis on monsoon-BC relationship, they should be singled out and cited. Please check if any.

Page 7. Line 6. What are the scaling factors in use? Previous analysis have shown BC emissions are biased low, and adjustment has be made to better agree with AAOD (Bond 2013) or radiative forcing (Xu et al., 2013) estimates from observations.

Xu, Y., R. Bahadur, C. Zhao, and L. R. Leung (2013), Estimating the radiative forcing of carbonaceous aerosols over California based on satellite and ground observations, Journal of Geophysical Research: Atmospheres, 118(19), 11148–11160, doi:10.1002/jgrd.50835. Bond, T. C., et al. (2013), Bounding the role of black carbon in the climate system: A scientific assessment,. J. Geophys. Res. Atmos., 118, 5380–5552

Line 11. Do you have one run for each configuration? Or there is an ensemble of runs?

Page 8. Line 26. Could you comment on which is more reliable (if using NCEP as the benchmark)?

Page 16. Line 24. "summary" -> "summarize"

Page 19. Since these two simulations are previously described in separated papers, it is worth noting if they are identical except for the meteorological field.

Fig 8 and Fig 9. Statistical test of the difference should be conducted and reflected in the figure.
* * *

---

## Referee Comment (RC3) · Anonymous Referee #1 · 3 Jul 2016

This is a nicely written manuscript. This study is useful to understand the linkage of climate circulation and Asia pollution. I only have a few minor comments:

1) The spatial resolution used in this study is relatively coarse, and some potential uncertainties related to it may be discussed.

2) How much confidence do the authors have in simulating surface BC using GEOS-Chem? How about vertical profiles? The authors intensively investigated the impact of monsoon on vertical changes of BCs, but the first question is whether GEOS-Chem is able to capture the vertical profile of BC? How many uncertainties can be inferred

from the potential bias of GEOS-Chem? 3) In the abstract, Lines 17-18, whether the differences between the weakest EASM and strongest EASM years are significant. In another word, by looking at the entire simulation period, the authors can get the mean and the standard deviation of BC. How does this change magnitude (i.e., 0.04-0.09, 0.03-0.04) compare to the 20-year variance? If we look at the inter-season variability (either summer or winter), do the weaker EASM always corresponds to higher BC whereas stronger EASM corresponds to lower BC?

4) Page 13, Line 22: the authors mentioned the effect due to non-China emissions. Maybe I missed something, I am not sure how the authors claim this impact is from non-China emissions.

5) Did the authors consider biogenic emissions in this study? If so, please add it. If not, please add the possible uncertainty due to this missing source.

6) In the abstract (Line 14), the authors mentioned that the differences in BCs are mainly due to the circulation. I am wondering whether there are any way to quantify the effect from circulation change, wet deposition, etc.

7) The layouts of section 3 and 4 are interesting. These two parallel sections went through similar figures twice (one for summer and one for winter). Not sure whether this is the best way to discuss, but I think the figures showing summer and winter together look good.

8) Page 4, Line 17: L. Wang et al., 2014 Please remove "L."

9) Figure 3. Is the spatial correlation significant? One way is to use lines or markers to indicate statistical significance, or mask the areas insignificant.
* * *

---

## Author Comment (AC1) · 1 Dec 2016

**We would like to thank the referee for the thoughtful and insightful comments. We have addressed all of the comments. Our responses are itemized below.**

*In this work authors attempted to study the impacts of the interannual variation of Eastern Asian summer and winter monsoon on variations of black carbon (BC) mass concentrations and direct radiative forcing (DRF) in Eastern China during 1986-2006. Overall this paper is quite lengthy and reads more like a technical report. The results presented in the paper solely rely on model simulations lack of any observational evidences or cross-validation with previous modeling studies of BC. Some issues with respect to the method descriptions sound vague. The clarification of these issues is critical to understand comprehensive results presented in this study. I recommend the major revision of the paper before the possible acceptance of ACP by addressing my following comments.*

*Major comments:*
*1. The methodology used in this study simply followed previous studies [Zhu et al., 2012; Yang et al. 2014]. That's fine. The results of BC are not surprising to me at all since BC is one of important fine aerosol types (i.e., PM2.5) discussed in Zhu et al. [2012]. It might be more interesting to emphasize the change of special characteristics of BC (e.g., whether or how the change of cloud layer between weakest and strongest Eastern Asian monsoon impacts on the BC absorption and DRF).*

Thanks for the suggestion. We compare differences in JJA (DJF) cloud fraction (%) between the five weakest and five strongest EASM (EAWM) years during 1986–2006. Plots are averaged over longitude range of 110–125 °E based on MERRA. Compared to the five strongest EASM years, larger cloud fraction exists in northern China and also above ~7 km in southern China in the five weakest monsoon years. For winter monsoon, we find increased cloud fraction in southern China but decreased cloud below ~5 km in northern China in the weakest monsoon years. However, the impact of changes in cloud layer due to the monsoon on BC DRF is not as significant as that of changes in BC distributions due to the monsoon.

[Figure]

Added discussions in Sect. 5 "It is also worth to point out that the BC DRF is also

dependent on factors such as cloud and background aerosol distributions (Samset et al., 2011), which can be influenced by the strength of the EAM (Liu et al., 2010; Zhu et al., 2012)…These aspects should be further investigated in future studies".

*2. The results presented in the paper solely rely on model simulations lack of any observational evidences or cross-validation with previous modeling studies of BC. This makes me wonder how the modeled BC mass concentrations in this work compare with historical observations available in Eastern China, especially during JJA and DJF (i.e., the seasons authors focus on in this work).*

Added discussions in Sect. 2.1 "We have systematically evaluated the BC simulations for 1980-2010 in China from the GEOS-Chem model (Li et al., 2016; Mao et al., 2016).

*3. Authors presented major results based on the difference between weakest and strongest Eastern Asia summer monsoon in Section 3 (covering Fig. 1a, Fig. 2a1, 2b1, Fig. 3a, . . .) and then from the difference in winter monsoon in Section 4 (covering Fig. 1b, Fig. 2a2,2b2, Fig. 3b, . . .). However, no discussions (linked to changes in winds or circulation patterns, etc) were made on the difference between summer and winter monsoon, which makes two sections sound like separate stories.*

Thanks for the suggestion. Added discussions in Sect. 5 "Different patterns of atmospheric circulation between summer and winter monsoon lead to the different distributions of BC in southern and northern China."…"In addition, the strength of the EAWM would influence the following summer monsoon via changes in the factors such as circulation and precipitation (e.g., Chen et al., 2000), and further affect the aerosols concentrations and radiative forcing. These aspects should be further investigated in future studies.".

*4. Majority results in this work (i.e., Fig. 4-12, Table 2-5) highly reply on the difference between weakest and strongest Eastern Asia summer monsoon (in Section 3). The selection of five weakest and strongest years in this work is slightly different with previous studies [Zhu et al., 2012; Yang et al. 2014] that used the same GEOS-4 met fields of 1986-2006 without any explanations. Please explain why authors choose different monsoon years as adopted in Zhu et al. 2012 and Yang et al. 2014.*

Added discussions "we examine the differences in the JJA mean surface BC concentrations between five weakest (1988, 1993, 1995, 1996, and 1998) and five strongest (1990, 1994, 1997, 2004, and 2006) EASM years during 1986–2006"… "We select these weakest (or strongest) monsoon years based on the five largest negative (or positive) values of the normalized EASMI in both GEOS-4 and MERRA

within 1986–2006. The selected monsoon years are thus slightly different with those from previous studies (Zhu et al., 2012; Yang et al. 2014) only based on GEOS-4 (weakest monsoon years (1988, 1989, 1996, 1998, and 2003), and strongest monsoon years (1990, 1994, 1997, 2002, and 2006))."

*5. On Page 5 Line 26, should 1980-2010 be 1986-2006, which overlaps the period between GEOS-4 and MERRA?*

Revised as suggested.

*6. On Page 8 Line 5-6, authors mentioned numerous studies have shown that the intensity of EAWM. . .. . .but they only cited one reference of Yan et al. (2009). It sounds contradictory.*

Added references "Guo et al., 1994; Ji et al., 1997; Chen et al., 2000; Jhun and Lee, 2004".

*7. In Section 3 and 4, authors enclosed values in the parenthesis but did not describe how they calculate these values, for instance the range of percentages on Page 8 Lines 22-23. Please add the clarification.*

Added clarification "the deviation from the mean (DM)".

*8. On Page 10 Lines 19-20, what is the cause of the different pattern of BC concentration between GEOS-4 and MERRA shown in Figure 5a?*

*Added discussions* "The different patterns of BC concentrations between GEOS-4 and MERRA in Fig. 5a are likely because of the different convection schemes used in the two meteorological data (Rienecker et al., 2011)."

*9. On Page 11 Lines 2-3, how does the convergence cause the increase in BC concentration and anticyclone wind pattern cause the decrease in BC concentration?*

Added discussions "Relative to the strongest EASM years, anomalous northerlies over northern China and anomalous northeasterlies over the western North Pacific in the weakest EASM years prevent the outflow of pollutants from northern China. In addition, southerly branch of the anomalous anticyclone in the south of the middle and lower reaches of the Yangtze River and nearby oceans strengthens the northward transport of aerosols from southern China to northern China.".

*10. On Page 11, Lines 8-10, I understand the convergence accompanied with the descending air prevents surface BC to the upper troposphere, causing the increase in surface BC. But I don't understand why the upward mass flux of BC also increases under the condition of convergence. Could you explain it?*

Added discussions "Compared to the strong monsoon years, the increased surface BC concentrations in northern China lead to higher upward mass fluxes of BC concentrations north of 25 °N in both MERRA and GEOS-4. In southern China, the lower surface BC concentrations in the weakest EASM years result in the decreased upward fluxes south of 25 °N. The pattern of the anomalous vertical transport of BC concentrations thus confirms the anomalous convergence in northern China and anomalous divergence in southern China in the weakest monsoon years."

*11. On Page 11 Line 13, please describe the method you calculate horizontal mass fluxes at the four lateral boundaries in details. Clearly, the net effect does not equal to the fluxes summed with values from four lateral boundaries. How do you calculate the net effect of horizontal mass fluxes over the specific region?*

The horizontal mass flues and the net effect is summarized in Table 3. Added details "The net effect is a larger inflow of BC by 1.01 (1.27 larger inflow + 2.40 larger inflow + 0.62 lower outflow − 3.28 larger outflow) kg s$^{-1}$ in GEOS-4 and 1.60 (1.01 larger inflow + 1.21 larger inflow + 0.67 lower outflow − 1.29 larger outflow) kg s$^{-1}$ in MERRA".

*12. On Page 11 Lines 23-25, why is there larger inflow at the east and north boundary and smaller outflow at the south and east boundary?*

Added discussions "The differences in winds between the weak and strong monsoon years lead to differences in horizontal transport of BC."

*13. On page 11 Lines 27-29, where do these two numbers (i.e., 0.09 and 0.27 kg/s) come from? Do you average them over the entire domain? Please specify the region your numbers are based on?*

Revised to "As a result, the weakest monsoon years in southern China have larger outflow fluxes of 0.09 (0.81 larger inflow − 0.91 larger outflow + 0.09 lower outflow − 0.08 lower inflow) and 0.27 (0.35 larger inflow − 0.72 larger outflow + 0.09 lower outflow + 0.01 larger inflow) kg s$^{-1}$ than the strongest monsoon years in GEOS-4 and in MERRA, respectively."

*14. On Page 13, Lines 4-5, could you clarify what is the direct radiative forcing efficiency of BC? BTW, did you notice that the shift of the center of the highest BC DRF from weakest to strongest? Could you explain what is the cause of the shift?*

Added clarification in the parentheses in Sect. 3.4 "radiative forcing exerted per gram of BC". Added discussions "We find largest BC-induced forcing at the latitude of 30–40 °N in the weakest monsoon years and 35–40 °N in the strongest monsoon years. The shift of the center of the highest BC DRF is likely due to the different vertical distributions of BC concentrations between the weakest and strongest monsoon years (**Fig. 5a**)."

*15. On Page 13, Lines 18-21, please add quantitative metrics to quantify the change of BC DRF in northern and southern China.*

Added in Table 4.

*16. On Page 13, Lines 26-27, how do you distinguish the DRF of BC between non-China emission and local sources? Did you offline run the radiative transfer model? If yes, please describe it in the section of method.*

*Added discussions in Sect. 2.1. "We also conduct simulation (VNOC) to quantify the contributions of the non-China emissions to BC. The configurations of the model simulation are the same as those in VMET, except that anthropogenic and biomass burning emissions in China are set to zero."*

*17. On Page 14, line 22, could you show PBLH in the supplement? Also explain how PBLH changes surface BC concentration.*

Now included the PBLH in Figure S2. Also added discussions "The lower PBLH in MERRA suppresses the convection and thus leads to higher BC concentrations in the surface ."

*18. On Page 15, Lines 23-24, what is the cause of the different response of BC concentration to the summer and winter monsoon in southern china?*

Added discussions in Sect. 5 "Different patterns of atmospheric circulation between summer and winter monsoon lead to the different distributions of BC in southern and northern China."

*19. On Page 17, lines 21-24. I cannot tell the lower column burden of tropospheric BC from your Figure 5b. It appears that the BC profile increase at all altitudes. Is it related to the change of clouds?*

Added discussions in parentheses "Figs. 5(b2) and 10(b2)" and also in the following paragraphs.

*20. On Page 17, Lines 24-27, why is DRF lower in the weakest monsoon years in southern china even though both BC surface concentration and column burden are higher, compared with the strongest monsoon years?*

The possible reasons are discussed in the following paragraphs.

*21. On Page 20, besides simply reporting what you conclude in this work, could you add some discussion about why eastern Asian summer and winter monsoon change BC concentration and DRF in northern and southern China differently? Is this difference important to contribute to the air quality regulation in different regions of China?*

Added discussions in Sect. 5 "Note that these different changes in BC concentrations and DRF between northern and southern China due to the EAM would be useful for proposing efficient air quality regulation in different regions of China."

*22. On Page 5 Lines 17-18, BC is assumed externally mixed with other aerosol species in this model. Could authors discuss the uncertainties of your results based on this assumption? How do results change if BC is partially internally or internally mixed with other aerosol species?*

Thanks for the suggestion. Added discussions in Sect. 3.4 "Note that the estimated DRF is associated with large uncertainties due to the BC mixing state used in model, which assumes external mixing of aerosols and gives a lower-bound estimate of BC DRF. Internal mixing of BC with scattering aerosols in the real atmosphere likely increases the estimates of DRF (e.g., Jacobson, 2001).".

*Minor comments:*
*1. Page 11 Line 12, change "summary" to "summarize".*

Revised.

2. *Figure 1, add the description of r31y and r21y.*

   Revised as "*r*_1980-2010" and "*r*_1986-2006".

3. *Figure 4, please move the row of a2 above b1 since you discussed a2 ahead of b1 in the context.*

   Revised.

4. *Figure 10, please label a1, a2, b1, and b2 in Figure.*

   Revised.

5. *Figure 12, How do you distinguish BC concentration attributed to non-China emissions and local China sources in the model? Please specify in the description of the model.*

Added discussions in Sect. 2.1: "We also conduct simulation (VNOC) to quantify the contributions of the non-China emissions to BC. The configurations of the model simulation are the same as those in VMET, except that anthropogenic and biomass burning emissions in China are set to zero.".

---

## Author Comment (AC2) · 1 Dec 2016

**We would like to thank the referee for the thoughtful and insightful comments. We have addressed all of the comments. Our responses are itemized below.**

*This study used a chemistry transport model to examine the impact of monsoon variability on black carbon distribution (surface concentration, vertical profile and direct forcing as an integrated measure) over China. Since the emissions are fixed in the model simulation, metrological variability is the dominant sources of pollution variability. The results of this study generally support many empirical analyses that linked observed pollution level with monsoon variability as extensively cited in the paper. I found the analysis is comprehensive and the interpretation of the results is convincing. I recommend publication after the following issues are addressed.*

*One of my main concerns is regard to the interpretation of summer monsoon impact. It is noted by the authors " differences in transport of BC due to the changes in atmospheric circulation are a dominant mechanism through which the EASM influences the variations of JJA BC". However, the role of precipitation in setting wet removal is not adequately tested (e.g. excessive rainfall in strong monsoon year can increase wet removal flux and also contribute to pollution reduction). Of course, I admit that precipitation can also be correlated to circulation and moisture transport, so a deeper question would be. How do you separate the effects of circulation (dispersion) vs. rainfall (removal)?*

Points well taken. Now the differences in wet deposition are included in Table 3 and the role of wet deposition is discussed in Sects. 3.3 and 4.3. "We also examine the impact of the changes in precipitation associated with the strength of the summer monsoon on BC concentrations, which is not as dominant as that of the winds. Compared to the strongest EASM years, increases in wet deposition of BC are found in the weakest monsoon years north of 28 °N in eastern China (Table 2), as a result of the high aerosol concentrations in the region and also the increased rainfall in the lower and middle reaches of the Yangtze River (around 30 °N). In the region south of 28 °N in eastern China, we find decreased wet deposition of BC in the weakest monsoon years because of the less rainfall and low BC concentrations in that region. " "Compared to the strongest EAWM years, enhanced wet deposition of BC are found in the weakest monsoon years in both northern and southern China (Table 2), likely because of the increased BC concentrations and precipitation in the corresponding regions."

*For wintertime, it is reasonable that circulation is the main cause due to lack of rainfall generally. However, in explaining the higher surface concentrations in weaker monsoon years, how do you separate the effects of (1) weaker horizontal transport that leads to higher total column loading of pollution buildup and (2) weaker vertical mixing that tends to put more pollution at the surface? Are these two processes working in the same direction or now? Which is more important in the*

*model?*

Included discussions in Sect. 4.3 "Compared to the strongest monsoon years, increases in upward mass flux of BC concentrations are found over 20–30 ° N and north of 40 ° N in the troposphere in the weakest monsoon years, confirming the increased surface BC concentrations in northern and southern China (**Figs. 4b** and **5b**)." "Weaker upward transport in the weakest monsoon years than the strongest years above 1-2 km in southern China (Fig. **7b**) is also not a dominate factor that contributes to the higher surface BC concentrations in the region (Tables 2 and 3)."

*How do you account for the warming trends in the simulation years? Are the temperature/ precipitation trends significant? Are they leading to trends in pollutions?*

Added discussions in Sect. 3.3. "We would like to point out that warming trend is not a significant factor to the variations of BC concentrations in the present study, as emissions are fixed at the 2010 levels and warming trend in the emissions are thus excluded. In addition, Yang et al. (2016) have systematically examined the trends of metrological parameters and $PM_{2.5}$ in eastern China for 1985–2005. They found positive trend in temperature and negative trend in precipitation while no significant trends in BC concentrations.".

*Minor points:*
*Page 1. Line 9. How do these affect "influence the variations of emissions"?*

*Added clarification in the* parentheses "biomass burning emissions*".*

*Page 3. Line 19."convention" -> "convection"*

Revised.

*Page 2. Line 24. This is just repeating the reference in page 1?*

Deleted.

*Page 4. Line 7. "matters" -> "matter"*

Revised.

*Page 5. Line 5. What's a more appropriate reference here than Ramanathan and Carmichael (2008) is Ramanathan and Xu (2010).*

Revised.

*Line 11. It is odd to compare 1980-2010 forcing over Asia with 1850-2005 forcing globally.*

Deleted.

*Line 14. "Changes in monsoon"? But you did not provide references that monsoon was indeed changing over EA.*

References now included.

*Line 22. If there is any statistical analysis on monsoon-BC relationship, they should be singled out and cited. Please check if any.*

Added discussions "Zhu et al. (2012) showed that simulated summer surface BC concentrations averaged over northern China (110–125 ° E, 28–45 ° N) are ~11% higher in the five weakest monsoon years than in the five strongest monsoon years for 1986–2006."

*Page 7. Line 6. What are the scaling factors in use? Previous analysis have shown BC emissions are biased low, and adjustment has be made to better agree with AAOD (Bond 2013) or radiative forcing (Xu et al., 2013) estimates from observations.*

Added discussions "We have systematically evaluated the BC simulations for 1980-2010 in China from the GEOS-Chem model (Li et al., 2016; Mao et al., 2016). We would like to point out that simulated BC concentrations are likely underestimated because of the biased low emissions (e.g., Bond et al., 2013; Xu et al., 2013; Mao et al., 2016) and coarse resolution of the model used. We discussed the adjustment of the biased low BC emissions using the scaling factor in our previous study by Mao et al. (2016). The adjustment of the BC emissions is not included in the present study, as we aim to discuss the impact of variations in meteorological parameters on BC. "

*Line 11. Do you have one run for each configuration? Or there is an ensemble of*

*runs?*

Only one ensemble run for the configurations. Revised to "More details about the anthropogenic and biomass burning emissions of BC are discussed by Mao et al. (2016).". "In each simulation, meteorological parameters are allowed to vary year to year, but anthropogenic and biomass burning emissions of BC are fixed at the year 2010 levels.".

*Page 8. Line 26. Could you comment on which is more reliable (if using NCEP as the benchmark)?*

Added discussions "MERRA is likely more reliable than the previous versions of GMAO metrological data products (e.g., GEOS-4 and GEOS-5), as MERRA has significant improved the convection and then precipitation and water vapor (Rienecker et al., 2011) by comparing to the reanalyses."

*Page 16. Line 24. "summary" -> "summarize"*

Revised.

*Page 19. Since these two simulations are previously described in separated papers, it is worth noting if they are identical except for the meteorological field.*

The details about the configurations of the two simulations are now included in Sect. 2.

*Fig 8 and Fig 9. Statistical test of the difference should be conducted and reflected in the figure.*

The statistical analysis are included in Table 4 and the related Sects.3.4 and 4.4.

---

## Author Comment (AC3) · 1 Dec 2016

**We would like to thank the referee for the thoughtful and insightful comments. We have addressed all of the comments. Our responses are itemized below.**

*This is a nicely written manuscript. This study is useful to understand the linkage of climate circulation and Asia pollution. I only have a few minor comments:*
*1) The spatial resolution used in this study is relatively coarse, and some potential uncertainties related to it may be discussed.*

Added discussions in Sect. 2.1 "We would like to point out that simulated BC concentrations are likely underestimated because of the .... and coarse resolution of the model used.".

*2) How much confidence do the authors have in simulating surface BC using GEOSChem? How about vertical profiles? The authors intensively investigated the impactof monsoon on vertical changes of BCs, but the first question is whether GEOS-Chemis able to capture the vertical profile of BC? How many uncertainties can be inferred from the potential bias of GEOS-Chem?*

Added discussions in Sect. 2.1 "We have systematically evaluated the BC simulations for 1980-2010 in China from the GEOS-Chem model (Li et al., 2016; Mao et al., 2016)." and in Sect. 3.4 "We would like to point out that few aircraft observations of BC vertical profile are available in China. Previous studies have evaluated the GEOS-Chem simulated vertical profiles of BC by using datasets from aircraft campaigns for the regions of the Northwest Pacific, North America, and the Arctic (Park et al., 2005; Drury et al., 2010; Wang et al., 2011).".

*3) In the abstract, Lines 17-18, whether the differences between the weakest EASM and strongest EASM years are significant. In another word, by looking at the entire simulation period, the authors can get the mean and the standard deviation of BC. How does this change magnitude (i.e., 0.04-0.09,0.03-0.04) compare to the 20-year variance? If we look at the inter-season variability (either summer or winter), do the weaker EASM always corresponds to higher BC whereas stronger EASM corresponds to lower BC?*

Thanks for the suggestions. Now mean and standard deviation of BC for 1986-2006 are included in Table 2 and the related discussions are in Sects. 3.3 and 4.3. "The difference in surface BC concentrations between the weakest and strongest summer monsoon years in each region is comparable or even larger than the corresponding standard deviation of JJA mean surface BC for 1986–2006 (Table 2).". "The difference in surface BC concentrations between the weakest and strongest winter monsoon years over each region is significant by comparing with the corresponding mean and standard deviation of DJF mean surface BC for 1986–2006 (Table 2)."

*4) Page 13, Line 22: the authors mentioned the effect due to non-China emissions. Maybe I missed something, I am not sure how the authors claim this impact is from non-China emissions.*

Added discussions in Sect. 2.1 "We also conduct simulation (VNOC) to quantify the contributions of the non-China emissions to BC. The configurations of the model simulation are the same as those in VMET, except that anthropogenic and biomass burning emissions in China are set to zero."

*5) Did the authors consider biogenic emissions in this study? If so, please add it. If not, please add the possible uncertainty due to this missing source.*

Added discussions in Sect. 2.1 "including both fossil fuel and biofuel emissions".

*6) In the abstract (Line 14), the authors mentioned that the differences in BCs are mainly due to the circulation. I am wondering whether there are any way to quantify the effect from circulation change, wet deposition, etc.*

Thanks for the suggestion. Now the effect from circulation change and wet deposition are included in Table 3 and the role of wet deposition is discussed in Sects. 3.3 and 4.3. "We also examine the impact of the changes in precipitation associated with the strength of the summer monsoon on BC concentrations, which is not as dominant as that of the winds. Compared to the strongest EASM years, increases in wet deposition of BC are found in the weakest monsoon years north of 28 °N in eastern China (Table 2), as a result of the high aerosol concentrations in the region and also the increased rainfall in the lower and middle reaches of the Yangtze River (around 30 °N). In the region south of 28 °N in eastern China, we find decreased wet deposition of BC in the weakest monsoon years because of the less rainfall and low BC concentrations in that region." "Compared to the strongest EAWM years, enhanced wet deposition of BC are found in the weakest monsoon years in both northern and southern China (Table 2), likely because of the increased BC concentrations and precipitation in the corresponding regions."

*7) The layouts of section 3 and 4 are interesting. These two parallel sections went through similar figures twice (one for summer and one for winter). Not sure whether this is the best way to discuss, but I think the figures showing summer and winter together look good.*

We would like to keep the Sect. 3 and 4 separately, which are likely readable and easy to follow.

*8) Page 4, Line 17: L. Wang et al., 2014 Please remove "L."*

Deleted.

*9) Figure 3. Is the spatial correlation significant? One way is to use lines or markers to indicate statistical significance, or mask the areas insignificant.*

Thanks for the suggestion. The dotted areas added in Figure 3 indicate statistical significance with 95% confidence from a two-tailed Student's t-test.

---

## Author Response (AR3)

**We would like to thank the referee for the thoughtful and insightful comments. We have addressed all of the comments. Our responses are itemized below.**

*The response appears sloppy. For example, the authors do not give the caption for the figure discussed in the 1st comment. In the 2nd comment, they add some references but not include references in the response. I have to go back to the revised manuscript to check what they are and whether the references are appropriate. Also, when they add some sentences in the revision, they do not give line numbers. Overall, the authors clarified most of my comments but there are still some unclear issues listed below. I recommend the minor revision of the paper before the possible acceptance of ACP by addressing my following comments.*

Thanks for the comments. We have now added the caption for the figure, references, and the line numbers in our responses.

*1) It is hard for me to understand your sentence filled with plus and minus sign in your response to my comments 11 and 13. Please try to interpret it using your own words and do not use math symbols.*

*Comments 11. On Page 11 Line 13, please describe the method you calculate horizontal mass fluxes at the four lateral boundaries in details. Clearly, the net effect does not equal to the fluxes summed with values from four lateral boundaries. How do you calculate the net effect of horizontal mass fluxes over the specific region?*

The horizontal mass fluxes and the net effects are summarized in Table 3. Revised to (P13L8-16) 'In northern China, the weakest (strongest) monsoon years in GEOS-4 show inflow fluxes of BC by 2.24 (0.97) kg s$^{-1}$ at the south boundary and by 6.60 (4.20) kg s$^{-1}$ at the west boundary, and outflow fluxes of BC by 3.44 (4.06) kg s$^{-1}$ at the north boundary and by 12.48 (9.20) kg s$^{-1}$ at the east boundary. The total effects are thus outflow fluxes by 7.08 kg s$^{-1}$ in the weakest monsoon years and by 8.09 kg s$^{-1}$ in the strongest monsoon years, resulting in a net effect of larger inflow of BC by 1.01 kg s$^{-1}$ in the weakest monsoon years than in the strongest monsoon years. Similarly, simulation results in MERRA show a net effect of larger inflow of BC by 1.60 kg s$^{-1}$ in the weakest monsoon years than in the strongest monsoon years.'

*Comments 13. On page 11 Lines 27-29, where do these two numbers (i.e., 0.09 and 0.27 kg/s) come from? Do you average them over the entire domain? Please specify the region your numbers are based on?*

The two numbers (i.e., 0.09 and 0.27 kg/s) are the differences in the net effect of the fluxes from four lateral boundaries in southern China (110–125 ° E, 20–27 ° N) between the weakest monsoon years and the strongest monsoon years. Revised to (P23L8-30) 'In southern China, we find inflow fluxes of BC by 0.62 (0.70) kg s$^{-1}$ at the south boundary and by 0.94 (0.13) kg s$^{-1}$ at the west boundary, and outflow fluxes

of BC by 1.79 (0.88) kg s$^{-1}$ at the north boundary and by 0.33 (0.42) kg s$^{-1}$ at the east boundary in the weakest (strongest) monsoon years in GEOS-4. The resulting effect is larger outflow fluxes of BC by 0.09 kg s$^{-1}$ in the weakest monsoon years (0.56 kg s$^{-1}$) than in the strongest monsoon years (0.47 kg s$^{-1}$). In MERRA, the weakest monsoon years also show larger outflow fluxes of BC by 0.27 kg s$^{-1}$, compared to the strongest monsoon years.'

*2) It is too vague to me in their response to my comment 18 when the authors ascribed different BC concentration to different patterns of atmospheric circulation due to winter and summer monsoon. Please specify the cause the different response of BC to summer and winter monsoon (e.g., how these two types of monsoon system differently affects BC source, sink and transport in southern China?).*

*Comments 18. On Page 15, Lines 23-24, what is the cause of the different response of BC concentration to the summer and winter monsoon in southern china?*

We discuss the atmospheric circulation via the fluxes at the four lateral boundaries in southern China. Added discussions in (P20L15-22) 'The net effect in southern China is a larger inflow of BC by 0.19 (0.40) kg s$^{-1}$ in the weakest EAWM years than in the strongest EAWM years, which leads to the higher BC concentrations in the weakest EAWM years. As we discussed in Sect. 3.3, the larger outflow fluxes of BC by 0.09 (0.27) kg s$^{-1}$ result in the lower BC concentrations in southern China in the weakest EASM years than in the strongest EASM years. Different patterns of atmospheric circulation between summer and winter monsoon thus lead to the different distributions of BC in southern China. '

*3) I cannot find the authors' discussions in response to my comments 19 and 20 in the revised manuscript. Please give line numbers!*

*Comments 19. On Page 17, lines 21-24. I cannot tell the lower column burden of tropospheric BC from your Figure 5b. It appears that the BC profile increase at all altitudes. Is it related to the change of clouds?*

We are discussing Figs. 5(b right) and 10(b2) here. We attribute the reduction in the DRF in the central China Plain largely to the decreased BC concentrations from 1–6 km, which likely because of the weaker vertical convection in the weakest monsoon years. Added "Figs. 5(b right) and 10(b2)" in parentheses (P21L18). (P22L13-17)'Relative to the strongest monsoon years, decreased BC concentrations are found in the weakest monsoon years …. from 1 to 6 km in the central China Plain. The decreased BC concentrations above 1–2 km lead to the reduction in the DRF in the two regions.'(P22L20-23) 'The lower concentrations above 1–2 km in the weakest monsoon years in southern China and the central China Plain are likely because of the weaker vertical convection at the corresponding altitudes in the weakest monsoon years than in the strongest monsoon years.'

*Comments 20. On Page 17, Lines 24-27, why is DRF lower in the weakest monsoon years in southern china even though both BC surface concentration and column burden are higher, compared with the strongest monsoon years?*

We attribute these discrepancies largely to the vertical distributions of BC concentrations. Added discussions (P21L22-23) 'We attribute these discrepancies largely to the vertical distributions of BC concentrations as discussed in the following paragraph.' (P22L13-17) 'Relative to the strongest monsoon years, decreased BC concentrations are found in the weakest monsoon years from 2 to 5 km in southern China …The decreased BC concentrations above 1–2 km lead to the reduction in the DRF in the two regions.'

**We would like to thank the referee for the thoughtful and insightful comments. We have addressed all of the comments. Our responses are itemized below.**

*This is a nicely written manuscript. This study is useful to understand the linkage of climate circulation and Asia pollution. I only have a few minor comments:*
*1)   The spatial resolution used in this study is relatively coarse, and some potential uncertainties related to it may be discussed.*

Added discussions in Sect. 2.1 (P8L6-8) "We would like to point out that simulated BC concentrations are likely underestimated because of the .... and coarse resolution of the model used.".

*2) How much confidence do the authors have in simulating surface BC using GEOSChem? How about vertical profiles? The authors intensively investigated the impactof monsoon on vertical changes of BCs, but the first question is whether GEOS-Chemis able to capture the vertical profile of BC? How many uncertainties can be inferred from the potential bias of GEOS-Chem?*

Added discussions in Sect. 2.1 (P8L4-6) "We have systematically evaluated the BC simulations for 1980-2010 in China from the GEOS-Chem model (Li et al., 2016; Mao et al., 2016)." and in Sect. 3.4 (P15L7-12) "We would like to point out that few aircraft observations of BC vertical profile are available in China. Previous studies have evaluated the GEOS-Chem simulated vertical profiles of BC by using datasets from aircraft campaigns for the regions of the Northwest Pacific, North America, and the Arctic (Park et al., 2005; Drury et al., 2010; Wang et al., 2011).".

*3) In the abstract, Lines 17-18, whether the differences between the weakest EASM and strongest EASM years are significant. In another word, by looking at the entire simulation period, the authors can get the mean and the standard deviation of BC. How does this change magnitude (i.e., 0.04-0.09,0.03-0.04) compare to the 20-year variance? If we look at the inter-season variability (either summer or winter), do the weaker EASM always corresponds to higher BC whereas stronger EASM corresponds to lower BC?*

Thanks for the suggestions. Now mean and standard deviation of BC for 1986-2006 are included in Table 2 and the related discussions are in Sects. 3.3 (P11L22-25) and 4.3 (P20L5-8). "The difference in surface BC concentrations between the weakest and strongest summer monsoon years in each region is comparable or even larger than the corresponding standard deviation of JJA mean surface BC for 1986–2006 (Table 2).". "The difference in surface BC concentrations between the weakest and strongest winter monsoon years over each region is significant by comparing with the corresponding mean and standard deviation of DJF mean surface BC for 1986–2006

(Table 2)."

*4) Page 13, Line 22: the authors mentioned the effect due to non-China emissions. Maybe I missed something, I am not sure how the authors claim this impact is from non-China emissions.*

Added discussions in Sect. 2.1 (P7L29-P8L2) "We also conduct simulation (VNOC) to quantify the contributions of the non-China emissions to BC. The configurations of the model simulation are the same as those in VMET, except that anthropogenic and biomass burning emissions in China are set to zero."

*5) Did the authors consider biogenic emissions in this study? If so, please add it. If not, please add the possible uncertainty due to this missing source.*

Added discussions in Sect. 2.1 (P7L12) "including both fossil fuel and biofuel emissions".

*6) In the abstract (Line 14), the authors mentioned that the differences in BCs are mainly due to the circulation. I am wondering whether there are any way to quantify the effect from circulation change, wet deposition, etc.*

Thanks for the suggestion. Now the effect from circulation change and wet deposition are included in Table 3 and the role of wet deposition is discussed in Sects. 3.3 (P14L10-18) and 4.3 (P20L26-29). "We also examine the impact of the changes in precipitation associated with the strength of the summer monsoon on BC concentrations, which is not as dominant as that of the winds. Compared to the strongest EASM years, increases in wet deposition of BC are found in the weakest monsoon years north of $28\,^{\circ}$N in eastern China (Table 2), as a result of the high aerosol concentrations in the region and also the increased rainfall in the lower and middle reaches of the Yangtze River (around $30\,^{\circ}$N). In the region south of $28\,^{\circ}$N in eastern China, we find decreased wet deposition of BC in the weakest monsoon years because of the less rainfall and low BC concentrations in that region." "Compared to the strongest EAWM years, enhanced wet deposition of BC are found in the weakest monsoon years in both northern and southern China (Table 2), likely because of the increased BC concentrations and precipitation in the corresponding regions."

*7) The layouts of section 3 and 4 are interesting. These two parallel sections went through similar figures twice (one for summer and one for winter). Not sure whether this is the best way to discuss, but I think the figures showing summer and winter together look good.*

We would like to keep the Sect. 3 and 4 separately, which are likely readable and easy

to follow.

Deleted.

*9) Figure 3. Is the spatial correlation significant? One way is to use lines or markers to indicate statistical significance, or mask the areas insignificant.*

Thanks for the suggestion. The dotted areas added in Figure 3 indicate statistical significance with 95% confidence from a two-tailed Student's t-test.

**We would like to thank the referee for the thoughtful and insightful comments. We have addressed all of the comments. Our responses are itemized below.**

*This study used a chemistry transport model to examine the impact of monsoon variability on black carbon distribution (surface concentration, vertical profile and direct forcing as an integrated measure) over China. Since the emissions are fixed in the model simulation, metrological variability is the dominant sources of pollution variability. The results of this study generally support many empirical analyses that linked observed pollution level with monsoon variability as extensively cited in the paper. I found the analysis is comprehensive and the interpretation of the results is convincing. I recommend publication after the following issues are addressed.*

*One of my main concerns is regard to the interpretation of summer monsoon impact. It is noted by the authors " differences in transport of BC due to the changes in atmospheric circulation are a dominant mechanism through which the EASM influences the variations of JJA BC". However, the role of precipitation in setting wet removal is not adequately tested (e.g. excessive rainfall in strong monsoon year can increase wet removal flux and also contribute to pollution reduction). Of course, I admit that precipitation can also be correlated to circulation and moisture transport, so a deeper question would be. How do you separate the effects of circulation (dispersion) vs. rainfall (removal)?*

Points well taken. Now the differences in wet deposition are included in Table 3 and the role of wet deposition is discussed in Sects. 3.3 (P14L10-18) and 4.3 (P20L26-29). "We also examine the impact of the changes in precipitation associated with the strength of the summer monsoon on BC concentrations, which is not as dominant as that of the winds. Compared to the strongest EASM years, increases in wet deposition of BC are found in the weakest monsoon years north of 28 °N in eastern China (Table 2), as a result of the high aerosol concentrations in the region and also the increased rainfall in the lower and middle reaches of the Yangtze River (around 30 °N). In the region south of 28 °N in eastern China, we find decreased wet deposition of BC in the weakest monsoon years because of the less rainfall and low BC concentrations in that region. " "Compared to the strongest EAWM years, enhanced wet deposition of BC are found in the weakest monsoon years in both northern and southern China (Table 2), likely because of the increased BC concentrations and precipitation in the corresponding regions."

*For wintertime, it is reasonable that circulation is the main cause due to lack of rainfall generally. However, in explaining the higher surface concentrations in weaker monsoon years, how do you separate the effects of (1) weaker horizontal transport that leads to higher total column loading of pollution buildup and (2) weaker vertical mixing that tends to put more pollution at the surface? Are these two processes working in the same direction or now? Which is more important in the*

*model?*

Included discussions in Sect. 4.3 (P19L27-30, P20L29-P21L2) "Compared to the strongest monsoon years, increases in upward mass flux of BC concentrations are found over 20–30 °N and north of 40 °N in the troposphere in the weakest monsoon years, confirming the increased surface BC concentrations in northern and southern China (**Figs. 4b** and **5b**)." "Weaker upward transport in the weakest monsoon years than the strongest years above 1-2 km in southern China (Fig. **7b**) is also not a dominate factor that contributes to the higher surface BC concentrations in the region (Tables 2 and 3)."

*How do you account for the warming trends in the simulation years? Are the temperature/ precipitation trends significant? Are they leading to trends in pollutions?*

Added discussions in Sect. 3.3 (P14L19-24). "We would like to point out that warming trend is not a significant factor to the variations of BC concentrations in the present study, as emissions are fixed at the 2010 levels and warming trend in the emissions are thus excluded. In addition, Yang et al. (2016) have systematically examined the trends of metrological parameters and $PM_{2.5}$ in eastern China for 1985–2005. They found positive trend in temperature and negative trend in precipitation while no significant trends in BC concentrations.".

*Minor points:*
*Page 1. Line 9. How do these affect "influence the variations of emissions"?*

*Added clarification in the* parentheses "biomass burning emissions*".*

*Page 3. Line 19."convention" -> "convection"*

Revised.

*Page 2. Line 24. This is just repeating the reference in page 1?*

Deleted.

*Page 4. Line 7. "matters" -> "matter"*

Revised.

*Page 5. Line 5. What's a more appropriate reference here than Ramanathan and Carmichael (2008) is Ramanathan and Xu (2010).*

Revised.

*Line 11. It is odd to compare 1980-2010 forcing over Asia with 1850-2005 forcing globally.*

Deleted.

*Line 14. "Changes in monsoon"? But you did not provide references that monsoon was indeed changing over EA.*

References now included. Added (e.g., Menon et al., 2002; Lau et al., 2006)

*Line 22. If there is any statistical analysis on monsoon-BC relationship, they should be singled out and cited. Please check if any.*

Added discussions (P5L27-29) "Zhu et al. (2012) showed that simulated summer surface BC concentrations averaged over northern China (110–125 °E, 28–45 °N) are ~11% higher in the five weakest monsoon years than in the five strongest monsoon years for 1986–2006."

*Page 7. Line 6. What are the scaling factors in use? Previous analysis have shown BC emissions are biased low, and adjustment has be made to better agree with AAOD (Bond 2013) or radiative forcing (Xu et al., 2013) estimates from observations.*

Added discussions (P8L4-12) "We have systematically evaluated the BC simulations for 1980-2010 in China from the GEOS-Chem model (Li et al., 2016; Mao et al., 2016). We would like to point out that simulated BC concentrations are likely underestimated because of the biased low emissions (e.g., Bond et al., 2013; Xu et al., 2013; Mao et al., 2016) and coarse resolution of the model used. We discussed the adjustment of the biased low BC emissions using the scaling factor in our previous study by Mao et al. (2016). The adjustment of the BC emissions is not included in the present study, as we aim to discuss the impact of variations in meteorological parameters on BC. "

*Line 11. Do you have one run for each configuration? Or there is an ensemble of runs?*

Only one ensemble run for the configurations. Revised to "More details about the anthropogenic and biomass burning emissions of BC are discussed by Mao et al. (2016).". "In each simulation, meteorological parameters are allowed to vary year to year, but anthropogenic and biomass burning emissions of BC are fixed at the year 2010 levels." (P7L19-21, 25-27).

*Page 8. Line 26. Could you comment on which is more reliable (if using NCEP as the benchmark)?*

Added discussions (P10L2-5) "MERRA is likely more reliable than the previous versions of GMAO metrological data products (e.g., GEOS-4 and GEOS-5), as MERRA has significant improved the convection and then precipitation and water vapor (Rienecker et al., 2011) by comparing to the reanalyses."

*Page 16. Line 24. "summary" -> "summarize"*

Revised.

*Page 19. Since these two simulations are previously described in separated papers, it is worth noting if they are identical except for the meteorological field.*

The details about the configurations of the two simulations are now included in Sect. 2. (P7L25-27)

*Fig 8 and Fig 9. Statistical test of the difference should be conducted and reflected in the figure.*

The statistical analysis are included in Table 4 and the related Sects.3.4 and 4.4.

References:

[revised manuscript text omitted]

Black carbon (BC) as a chemically inert species is a good tracer to investigate the impact of the meteorological parameters and the EAM on the interannual variations of aerosols. BC is an important short-lived aerosol; the reduction of BC emissions is identified as a near-term approach to benefit the human health, air quality, and climate change efficiently (Ramanathan and Xu, 2010Ramanathan and Carmichael, 2008; Shindell et al., 2012; Bond et al., 2013; IPCC, 2013; Smith et al., 2013). BC emissions in China have been dramatically increased in the recent several decades, which contribute about 25% of the global total emissions (Cooke et al., 1999; Bond et al., 2004; Lu et al., 2011; Qin and Xie, 2012; Wang et al., 2012). Observed annual mean surface BC concentrations are typically about 2–5 $\mu$g m$^{-3}$ at rural sites (Zhang et al., 2008). Simulated annual direct radiative forcing (DRF) due to BC at the top of the atmosphere (TOA) is in the range of 0.58–1.46 W m$^{-2}$ in China, reported by previous modeling studies (summarized in Li et al., 2016). Mao et al. (2016) using the GEOS-Chem model showed that annual mean BC DRF averaged over China increases by 0.35 W m$^{-2}$ (51%) between 2010 and 1980, which is comparable to the global annual mean DRF values of BC (0.4 W m$^{-2}$), tropospheric ozone (0.4 W m$^{-2}$), and carbon dioxide (1.82 W m$^{-2}$) (IPCC, 2013).

[revised manuscript text omitted]